# ANY3D-VLA: Enhancing VLA Robustness via Diverse Point Clouds

**Xianzhe Fan** [1 2]   **Shengliang Deng** [1 2]   **Xiaoyang Wu** [1]   **Yuxiang Lu** [1]   **Zhuoling Li** [1]   **Mi Yan** [2 3]   **Yujia Zhang** [1]
**Zhizheng Zhang** [2]   **He Wang** [2 3]   **Hengshuang Zhao** [1]

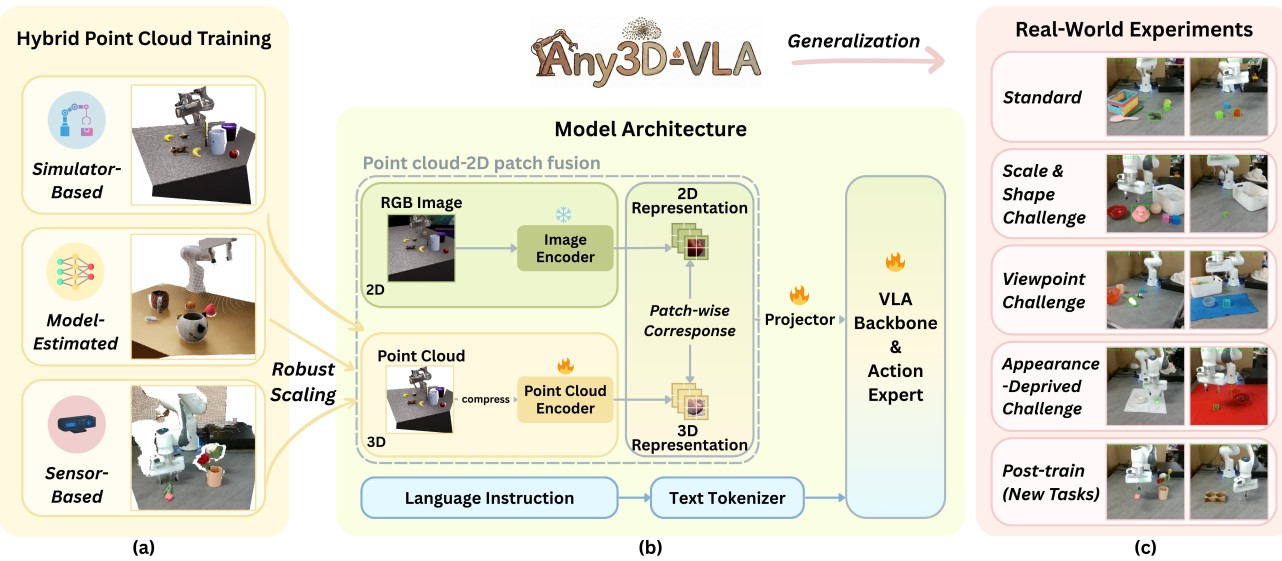

*Figure 1.* We propose ANY3D-VLA. It unifies simulator, sensor, and model-estimated point clouds in the training pipeline **(a)**, enabling diverse inputs and learning domain-agnostic 3D representations that are fused with the corresponding 2D representations **(b)**. **(c)** shows our experimental results in real-world settings.

## Abstract

Existing Vision-Language-Action (VLA) models typically take 2D images as visual input, which limits their spatial understanding in complex scenes. How can we incorporate 3D information to enhance VLA capabilities? We conduct a pilot study across different observation spaces and visual representations. The results show that explicitly lifting visual input into point clouds yields representations that better complement their corresponding 2D representations. To address the challenges of (1) scarce 3D data and (2) the domain gap induced by cross-environment differences and

depth-scale biases, we propose ANY3D-VLA. It unifies the simulator, sensor, and model-estimated point clouds within a training pipeline, constructs diverse inputs, and learns domain-agnostic 3D representations that are fused with the corresponding 2D representations. Simulation and real-world experiments demonstrate ANY3D-VLA's advantages in improving performance and mitigating the domain gap. Our project homepage is available at https://xianzhefan.github.io/Any3D-VLA.github.io.

## 1. Introduction

Vision-Language-Action (VLA) models, trained on massive collections of action trajectories paired with language instructions, hold great promise for achieving general-purpose embodied intelligence (Kim et al., 2025b; Deng et al., 2025; Black et al., 2025b;a; Bu et al., 2025). While these models are increasingly powerful in language and semantic understanding, their spatial understanding is still largely inherited

[1]School of Computing and Data Science, The University of Hong Kong, Hong Kong SAR, China [2]Galbot, Beijing, China [3]Peking University, Beijing, China. Correspondence to: Xianzhe Fan <xianzhef@connect.hku.hk>, Hengshuang Zhao <hszhao@cs.hku.hk>.

*Proceedings of the 43rd International Conference on Machine Learning*, Seoul, South Korea. PMLR 306, 2026. Copyright 2026 by the author(s).

from 2D visual backbones. As a result, they are particularly brittle in scenarios involving small objects, viewpoint changes, and occlusions (Qu et al., 2025).

To enhance spatial understanding, the community has explored multiple ways to inject 3D information into VLAs. Some works, while keeping the input strictly 2D, introduce a depth-pretrained encoder (Yuan et al., 2025) or a pretrained spatial encoder (Lin et al., 2025b; Li et al., 2025b) (e.g., VGGT (Wang et al., 2025)) to incorporate depth and 3D priors into the model. Other approaches enhance spatial perception by taking depth maps as input (Qu et al., 2025; Bhat et al., 2025; Shi et al., 2025a; Li et al., 2025e). Several studies investigate how to incorporate point-cloud information (Li et al., 2025a;d; Sun et al., 2025; Chen et al., 2025a; Singh et al., 2025), but they are still constrained by the lack of a pretrained point encoder or by relying on non-parametric 3D tokenizers. Moreover, some methods use only point-cloud inputs (Singh et al., 2025), or treat point clouds relatively independently from 2D inputs (Li et al., 2025a).

How can we incorporate 3D information to enhance VLA capabilities? We conduct a pilot study on various observation spaces and visual representations. The results indicate that while implicit and reconstruction-based spatial priors injected via models like VGGT improve visual representations, they remain imprecise when handling fine-grained spatial relationships. In contrast, representations obtained by explicitly projecting visual inputs into point-cloud space can more effectively complement 2D representations. However, 3D VLAs still face bottlenecks in scalable training and real deployment: (1) compared to the massive amount of 2D image data, 3D data is extremely scarce; (2) 3D data from different environments (e.g., simulators, sensors) differs substantially in noise characteristics, scale distributions, and geometric biases; (3) high-quality depth inputs are often required, such as high-precision depth hardwares. These factors limit the model's ability to generalize across environments. Motivated by this, we explore a new paradigm: incorporating diverse point cloud sources (simulator, sensor, and model-estimated) into training.

We propose ANY3D-VLA, a plug-in pipeline for existing VLA backbones (Figure 1). Given RGB images with optional depth, we first lift the visual input to point clouds and compress them. A pre-trained point cloud encoder then produces point-wise embeddings. Finally, we align and fuse the 3D embeddings with corresponding 2D patch features, and feed the fused representation into the downstream backbone. We synthesize a large-scale RGBD pre-training dataset for VLA tasks, covering diverse depth sources to construct varied point-cloud inputs. In real-world deployment, ANY3D-VLA does not rely on expensive depth hardware or stringent data-collection conditions; with hybrid point cloud training, it learns more robust 3D geometric reasoning.

We conduct zero-shot evaluations in the real world under a range of perturbations (e.g., intra-class scale/shape variations and viewpoint perturbations), and fine-tune on new tasks (e.g., flower arrangement) using a small amount of real-world demonstration data. ANY3D-VLA is tested with two input settings: sensor-based and model-estimated point clouds. In zero-shot evaluation, ANY3D-VLA achieves a maximum overall accuracy of 62.5%, outperforming the best baseline by 29.2%; in post-training, the maximum overall accuracy reaches 93.3%. Thanks to the point-cloud–2D fused representation and the hybrid point-cloud training strategy, the model achieves robust sim-to-real generalization even when depth quality and scale vary. To further validate generalization in standardized environments, we also report results on LIBERO and CALVIN benchmarks.

The contributions of this paper are summarized as follows: (1) We propose ANY3D-VLA. By lifting visual inputs into point clouds, compressing them, and fusing 2D–3D representations, our method provides a general and modular framework for injecting 3D representations into VLAs. (2) To address the scaling bottlenecks of 3D VLA training and the cross-environment domain gap, we introduce a hybrid point-cloud training strategy and construct a large-scale RGBD dataset for VLA tasks. (3) We conduct extensive evaluations in simulation and real-world. Results show that ANY3D-VLA achieves superior performance across various complex scenarios. It remains robust even when depth inputs are noisy or exhibit scale bias during deployment.

## 2. Related Work

**Vision-Language-Action Models.** VLAs inject robot control action tokens on top of a vision-language backbone, unifying perception, language understanding, and action generation within a single model (Ghosh et al., 2024; Pertsch et al., 2025; Black et al., 2025b;a; Shi et al., 2025b; Kim et al., 2025a; Hou et al., 2025; Zhao et al., 2025). Recent works typically adopt transformer-based architectures that predict actions autoregressively and are trained on large-scale robot trajectories.

Most VLAs make decisions primarily based on 2D images and make limited use of 3D geometry, which leads to shortcomings in spatial understanding. To address this, a growing body of work has explored ways to incorporate 3D information (3D VLAs). Some approaches leverage spatial priors learned during training and require only RGB inputs at inference time, without explicitly providing depth maps or point clouds (Lin et al., 2025b; Yuan et al., 2025). Spatial Forcing (Li et al., 2025b) aligns intermediate visual embeddings of VLAs with geometric representations from a pre-trained 3D foundation model (VGGT). DepthVLA (Yuan et al., 2025) employs a depth expert to enhance the model's spatial understanding. Other methods directly incorporate depth maps

to improve spatial perception (Qu et al., 2025; Bhat et al., 2025; Shi et al., 2025a; Li et al., 2025e). 3D-CAVLA (Bhat et al., 2025) integrates chain-of-thought-style task descriptions, depth embeddings, and region-of-interest pooling to improve scene awareness. There are also efforts to introduce point-cloud information (Li et al., 2025d; Sun et al., 2025; Chen et al., 2025a; Singh et al., 2025). PointVLA (Li et al., 2025a) injects point-cloud features into a pre-trained VLA action expert via an injector, without retraining the entire model. 3DS-VLA (Li et al., 2025d) encodes 3D spatial observations using a pre-trained 2D vision-language model and establishes 3D spatial constraints to facilitate spatiotemporal reasoning. However, many of these methods are limited by non-pretrained point encoders or non-parametric 3D tokenizers; some rely solely on point-cloud inputs (Singh et al., 2025) or treat point clouds relatively independently from 2D inputs (Li et al., 2025a).

**Simulator, Sensor-Based, and Model-Estimated Depth for Point-Cloud Construction.** Using camera intrinsics, an RGB image with a corresponding depth map can be lifted into a point cloud in the camera coordinate frame. Large-scale simulation can synthesize high-quality metric depth, whereas real robots often rely on consumer-grade depth sensors (Li et al., 2025a;d;c) or depth estimation (Li et al., 2025e; Liu et al., 2025). Depth estimation mainly follows two approaches: (1) using single-frame or multi-frame depth estimation models (*e.g.*, UniDepthV2 (Piccinelli et al., 2025), Depth Anything 3 (Lin et al., 2025a)) to directly predict metric depth; (2) using feed-forward 3D geometry models (*e.g.*, MapAnything (Keetha et al., 2025)) to regress camera parameters and scene geometry from RGB images, and then computing metric depth via a camera model. Notably, sensor depth and simulator depth differ substantially in noise characteristics and quality, which can easily introduce a sim-to-real gap. Meanwhile, model-estimated depth may suffer from scale drift due to camera-parameter estimation errors, lighting changes, or viewpoint perturbations, thereby reducing accuracy. Nevertheless, these depth signals can still provide critical spatial cues to varying degrees, and RGB-based depth estimation also offers a pathway for large-scale data acquisition and scaling.

## 3. Dataset and Benchmark

We synthesized an RGBD dataset at scale in a simulator. The dataset is built from the Objaverse LVIS subset (Deitke et al., 2023), selecting 290 categories and 10,680 instances. In each episode, we randomly generate a cluttered object layout on a $40\text{cm} \times 50\text{cm}$ tabletop region, using physically valid poses consistent with the category semantics. Expert trajectories are produced by generating candidate grasp poses with BoDex (Chen et al., 2025b), performing one-shot collision-avoidance trajectory planning with CuRobo

(Sundaralingam et al., 2023), and verifying executability in MuJoCo (Todorov et al., 2012). Visual rendering is performed in Isaac Sim (Mittal et al., 2023): we randomize lighting, materials, backgrounds, and camera extrinsics, and render images from a single viewpoint. The camera intrinsics are matched to those of a RealSense D435. At each time step, depth maps aligned with RGB pixels are provided, acquired via two methods: (1) directly exporting depth using the Isaac Sim rendering pipeline to ensure strict alignment with the RGB images under the same camera model; and (2) estimating metric depth from RGB using a model and resizing it to a unified resolution. Depth is stored as a single-channel float32 image with shape $256 \times 256 \times 1$. Invalid or out-of-range pixels are set to 0.

To validate the effectiveness of pre-training in **simulation**, we constructed an RGBD evaluation dataset as a benchmark using the same procedure. This dataset includes 15 object categories that appeared in the pre-training data, while the layouts and backgrounds are randomly generated and unseen during pre-training, resulting in 95 distinct scenes. Figure 5 (Appendix B) shows examples from our large-scale synthetic pre-training RGBD dataset.

## 4. Pilot Study: Different Observation Spaces and Visual Representations

Let $s_t \in \mathcal{S}$ denote the world state at time $t$. An observation function $h$ maps $s_t$ to an observation $o_t = h(s_t)$, where $o_t \in \mathcal{O}$. Conditioned on the observation $o_t$ and an instruction $\tau$, the policy produces an action $a_t$. In our setting, $h$ collects RGB images, optional depth, and proprioceptive states. For VLA tasks, we compare five different approaches: all start from the same world state $s_t$ but construct observations $o_t$ through different perception and processing pipelines, differing in how explicitly geometric information is provided and how it is represented. The five observation-space designs and visual representations are summarized in Table 1 and Figure 7, with details provided in Appendix C.

To isolate the impact of observation space design and visual representation construction on VLA performance, we adopt the following controlled settings: (1) We use the same benchmark in simulation (§3): both training and evaluation rely on the simulator's ground-truth metric depth and a single-view configuration, eliminating the influence of noise and scale bias. (2) We uniformly freeze the image encoder and only fine-tune the last four layers of the other branch (if present). Across the following four settings, the fusion layer is kept identical: we concatenate the two feature streams along the channel dimension and apply a linear projection to align them to the DINOv2+SigLIP feature dimension. (3) Aside from the visual module, all methods share the same model architecture and training strategy.

*Table 1.* Summary of the five settings. Let $v$ be the view index, denoting the observation from the $v$-th camera (out of $V$ views). Let $I_t^{(v)}$ denote the RGB image captured at time $t$ from view $v$, and $D_t^{(v)}$ denote the depth map from view $v$.

| Setting | Observation space | Depth used? | 3D explicit? | Visual module (encoders/representation) |
|---|---|---|---|---|
| 2D-only | $o_t^{\text{rgb}} = \{I_t^{(v)}\}_v$ | $\times$ | $\times$ | Standard image encoder on RGB (DINOv2+SigLIP); outputs 2D patch tokens only. |
| Implicit-depth RGB | $o_t^{\text{impl-depth}} = \{I_t^{(v)}\}_v$ | $\times$ | $\times$ | Two-branch encoders: (1) standard image encoder (DINOv2+SigLIP), (2) depth-pretrained image encoder (Depth Anything v2 encoder; DPT decoder discarded). Concatenate patch-level tokens. |
| Implicit-3D RGB | $o_t^{\text{impl-3d}} = \{I_t^{(v)}\}_v$ | $\times$ | $\times$ | Two-branch encoders: (1) standard image encoder (DINOv2+SigLIP), (2) spatial foundation model encoder (VGGT) trained with multiview geometric reconstruction objectives; geometry heads discarded. Concatenate patch-level tokens. |
| RGBD image-plane | $o_t^{\text{rgbd}} = \{[I_t^{(v)}, D_t^{(v)}]\}_v$ | $\checkmark$ | $\times$ | Explicit depth as image-plane channels: feed $I_t^{(v)}$ and $D_t^{(v)}$ into the **same** image encoder (DINOv2+SigLIP) to obtain RGB/depth tokens in the image plane; fuse at the patch level. |
| Point cloud–2D patch fusion | $o_t^{\text{pc}} : \{[I_t^{(v)}, D_t^{(v)}]\}_v \to \mathcal{P}_t$ | $\checkmark$ | $\checkmark$ | Lift RGBD to point cloud $\mathcal{P}_t = \{(x_i, y_i, z_i, c_i, n_i)\}_i$ using intrinsics; 3D-space compression + pretrained point-cloud encoder (Concerto); align to 2D patches and fuse with 2D image tokens (DINOv2+SigLIP). |

*Table 2.* Comparison of different observation spaces and visual representations in the simulator. 'Single-Trial SR' denotes success on the first attempt, 'Test SR' within three attempts, and 'Grasp SR' for grasping any object.

| Method | Single-Trial SR (%) | Test SR (%) | Grasp SR (%) |
|---|---|---|---|
| 2D-only | 45.3 | 72.6 | 80.0 |
| Implicit-depth RGB | 55.8 | 78.9 | 85.3 |
| Implicit-3D RGB | 46.3 | 78.9 | 87.4 |
| RGBD image-plane | 56.8 | 76.8 | 87.4 |
| Point cloud–2D patch fusion | **61.1** | **80.0** | **89.5** |

As shown in Table 2, the point cloud–2D patch fusion paradigm achieves the best performance, with the most pronounced advantage in Single-Trial SR: compared to the second-best result of 56.8%, it improves by 4.3%, indicating a higher probability of completing the task on the first attempt. Notably, even in simulation with perfect depth, directly feeding depth as an additional image channel, or injecting implicit and reconstruction-based spatial priors via models such as VGGT, yields relatively limited gains. The potential reason for this is that implicit methods (such as VGGT or depth-pretrained encoders) rely on reconstruction objectives to learn spatial features; consequently, they often lack precise metric alignment and are prone to spatial hallucinations during fine-grained manipulation. Conversely, regarding methods that directly input depth maps: while they introduce explicit depth, treating it as a 2D channel input compromises the inherent topological structure of the 3D data. 2D backbones struggle to effectively infer occlusion relationships and absolute scales from flattened depth maps. This suggests that the key to performance improvements is not whether depth is provided, but how the geometric information carried by depth is represented and exploited. Rather than encoding geometric cues on the image plane, fusing native sparse 3D structure (*i.e.*, compressed point clouds) with the corresponding 2D patch representations provides more direct and more stable spatial constraints and interaction geometry required for manipulation.

## 5. ANY3D-VLA

We propose ANY3D-VLA, which adopts a point cloud–2D patch fusion approach (inspired by §4) along with a hybrid point cloud training strategy.

### 5.1. Overall Architecture

ANY3D-VLA integrates a Vision-Language Model (VLM) with an action expert (Black et al., 2025b), and connects them via a Progressive Action Generation (PAG) mechanism (Deng et al., 2025). The VLM comprises a trainable large language model InternLM2 1.8B (Cai et al., 2024), a visual observation module (§5.2), and a trainable projector that maps visual representations into the language space. We use a conditional flow-matching action expert (Lipman et al., 2023) to generate fine-grained end-effector actions.

### 5.2. Visual Observation Module

**Point-Cloud Construction and 3D Compression.** Given camera intrinsics $(f_x, f_y, c_x, c_y)$, the 3D point of a pixel $(u, v)$ with metric depth $d$ in the camera coordinate frame is

$$x = \frac{(u - c_x)d}{f_x}, \quad y = \frac{(v - c_y)d}{f_y}, \quad z = d. \quad (1)$$

We apply this conversion to all valid depth pixels. To avoid the computational cost of feeding all points directly into the encoder, we perform in $(x, y, z)$ space the same grid sampling as Sonata (Wu et al., 2025): we partition the workspace into fixed-resolution 3D grid cells, aggregate points that fall into the same cell, and keep only one representative point per cell, yielding a more compact and spatially more uniform 3D grid representation. We refer to this procedure as *3D compression*. Details are provided in Appendix D.

**Vision Encoder.** The compressed point cloud is fed into a pre-trained point cloud encoder $E_{3D}$. In our experiments, we employ Concerto (Zhang et al., 2025; 2026), which is pre-trained on large-scale 2D and 3D data. The encoder takes point coordinates $\mathbf{x}_i \in \mathbb{R}^3$, together with per-point attributes $\mathbf{q}_i$ (color and normal) as input, and outputs point-wise features $\{\mathbf{f}_i^{3D}\}_{i=1}^N = E_{3D}(\{(\mathbf{x}_i, \mathbf{q}_i)\}_{i=1}^N)$. We freeze most parameters of the encoder and fine-tune only the last few sparse convolution layers. In parallel, we use an image encoder $E_{2D}$ to map RGB images into patch-level tokens. The encoder can be instantiated with models such as DINOv2 (Oquab et al., 2024) and SigLIP (Zhai et al., 2023), which jointly capture semantic and geometric details (Karamcheti et al., 2024). Implementation details are provided in Appendix E.

**Patch-Wise Alignment.** To align and project 3D features onto 2D patches, we assign each 3D point $p_i$ a ViT patch index $a_i$. Concretely, we first project the 3D point back onto the image plane to obtain $(u_i, v_i) = \pi(\mathbf{x}_i)$, discretize the image plane into a regular patch grid, and compute the corresponding linear patch index $a_i$. We define $\mathcal{P}_j = \{ i \mid a_i = j \}$ as the set of points assigned to the $j$-th patch, and perform scatter-mean aggregation (Zaheer et al., 2017) over point features to obtain the patch-level 3D feature:

$$\mathbf{g}_j^{3D} = \begin{cases} \frac{1}{|\mathcal{P}_j|} \sum_{i \in \mathcal{P}_j} \mathbf{f}_i^{3D}, & |\mathcal{P}_j| > 0, \\ \mathbf{e}^{3D}, & |\mathcal{P}_j| = 0. \end{cases} \quad (2)$$

When no point falls into patch $j$, we use a learnable empty token $\mathbf{e}^{3D}$ as the patch-level 3D feature $\mathbf{g}_j^{3D}$. This patch-level alignment is a core component of our design: we perform native 3D understanding within the camera coordinate system and then project it back onto the 2D patch, rather than merely encoding depth on the image plane.

**2D–3D Fusion.** We first project the patch-level 3D feature $\mathbf{g}_j^{3D}$ to the token dimension via a learnable linear layer, yielding $\mathbf{h}_j^{3D} = W_{3D}\mathbf{g}_j^{3D}$. For each patch $j$, we have a 2D visual token $\mathbf{h}_j^{2D}$ and a corresponding 3D token $\mathbf{h}_j^{3D}$. We concatenate the 2D and 3D tokens and pass the result through a small MLP to obtain a residual vector $\delta_j = \text{MLP}([\mathbf{h}_j^{2D}; \mathbf{h}_j^{3D}])$. To avoid destroying the pre-trained 2D representation, we adopt a gated residual fusion:

$$\mathbf{h}_j^{\text{fused}} = \mathbf{h}_j^{2D} + \sigma(g) \cdot \text{LayerNorm}(\delta_j), \quad (3)$$

where $g$ is a learnable scalar gating parameter (initialized to -2.1972 so that $\sigma(g)$ is small at the beginning of training), and $\sigma(\cdot)$ is the sigmoid function. The fused token sequence $\{\mathbf{h}_j^{\text{fused}}\}$ from all views, together with the language tokens and proprioceptive tokens, is then fed into a VLA backbone. This design treats 3D representations as *corrections* to 2D representations, rather than replacing them entirely.

## 5.3. Training Strategy

We follow GraspVLA's imitation learning and PAG training paradigm (Deng et al., 2025). The model takes as input image observations and the corresponding point clouds, the language instruction, and proprioceptive data. The VLM autoregressively predicts intermediate discrete tokens, after which a conditional flow-matching action expert generates continuous end-effector action chunks. Training details of ANY3D-VLA are provided in Appendix F.

**Loss Function.** We jointly optimize the VLM head and the flow-matching action expert. For each batch, we simultaneously sample from the internet grounding dataset GRIT (Peng et al., 2023) and our synthetic RGBD dataset: the former is only used to supervise the VLM to autoregressively predict bounding-box tokens of the object to be grasped, while the latter additionally supervises the prediction of grasp pose tokens and the end-effector action trajectory via flow matching. We do not incorporate any explicit reconstruction losses for depth or point clouds, aiming to demonstrate that the performance gains stem primarily from superior spatial observation and representation rather than auxiliary supervision. The full form of the loss function is provided in Appendix F.1.

**Hybrid Point Cloud Training.** To account for variations in depth quality across different sources during actual deployment, we design three training settings. **Setting 1 (Simulator Only)**: All training samples use the simulator ground-truth metric point cloud. **Setting 2 (Hybrid Point Cloud)**: The model is trained *from scratch* on a dataset with hybrid sources. Specifically, for each recorded trajectory, we select a source with a fixed probability $p$: either the simulator point cloud (or the sensor-based point cloud for real-world datasets), or the metric point cloud estimated from a single RGB frame. Mixing ratios and visualizations of depth differences are provided in Table 7 and Figure 8 (Appendix F.3). **Setting 3 (Sensor Only)**: For real-world post-training datasets (§6.1.3), all training samples use sensor-based point cloud. For Setting 2, the model is exposed to all point-cloud types throughout training, encouraging the 3D encoder and fusion layers to learn geometric patterns that are invariant to depth sources, thereby tolerating scale bias and other imperfections in model-estimated point clouds.

## 6. Experiments Centered on ANY3D-VLA

We conduct a series of experiments around ANY3D-VLA to evaluate its performance (1) in real-world and simulation settings, (2) under different point-cloud sources, and (3) in few-shot adaptation with limited real-world data.

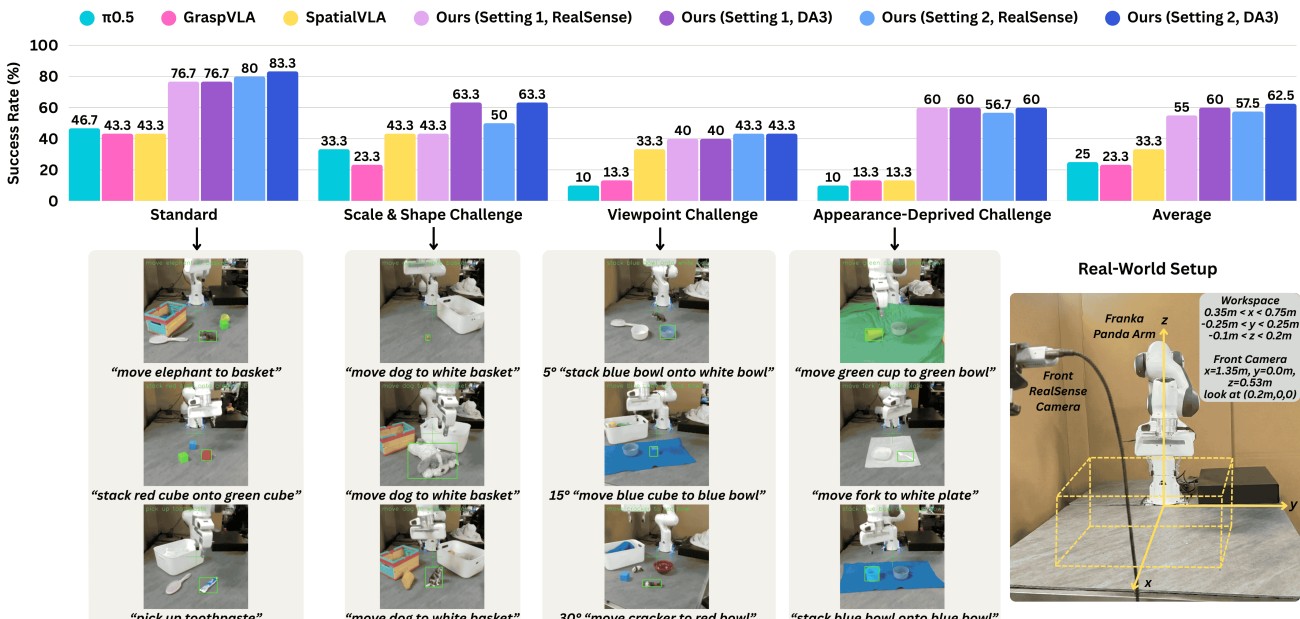

*Figure 2.* Zero-shot comparisons in the real world. For the training dataset, *Setting 1* utilizes only the simulator point cloud, whereas *Setting 2* incorporates both the simulator and multiple model-estimated point clouds (§5.3). During inference, *RealSense* refers to the sensor-based point cloud, while *DA3* refers to the point cloud derived from Depth Anything 3 depth predictions. For instance, (Setting 1, RealSense) denotes training on pure simulator point clouds and inferring with RealSense sensor point clouds.

## 6.1. Real-World Experiments

### 6.1.1. REAL-WORLD SETUP

**Baseline Models.** We select $\pi_{0.5}$ (Black et al., 2025a) and GraspVLA (Deng et al., 2025) as 2D VLA baselines, and SpatialVLA (Qu et al., 2025) as a 3D VLA baseline. All models are pre-trained on the **full monocular simulated dataset** and use the **same action chunking** method (Zhao et al., 2023), with the chunk length set to 4. Training hyperparameters for each model are provided in Table 9 (Appendix H). We evaluate the models in simulation, training until the success rate converges, and then select the best-performing checkpoint for real-world testing.

**Point Cloud Estimator Selection.** To ensure the reliability of real-world deployment, we first verify that the policy trained solely on simulator point clouds remains robust when exposed to relatively coarse model-estimated point clouds. As shown in Table 8 (Appendix G), we evaluate the model in the simulation environment by fixing the RGB input while utilizing point clouds generated by different models. The results demonstrate that our policy maintains competitive performance even with imperfectly estimated point clouds, effectively validating the feasibility of directly using model-estimated point clouds in the real world. We jointly consider prediction accuracy and inference latency, and conduct a qualitative comparison via point-cloud visualizations, ultimately selecting Depth Anything 3 (Lin

et al., 2025a) for real-world experiments. Real-world deployment details (hardware, camera setup, and workspace) are provided in Appendix I.

### 6.1.2. ZERO-SHOT COMPARISONS IN THE REAL WORLD

To evaluate ANY3D-VLA's zero-shot generalization ability and robustness in the real world, we design four challenging test sets: (1) **Standard:** Relatively simple scenes, with no more than six objects on the tabletop, and target objects mostly of conventional shapes and scales. (2) **Scale & Shape Challenge:** Scenes with substantial intra-class variations in size and shape, *e.g.*, dogs and bottles of different sizes and appearances; this set also includes geometrically challenging target objects, such as elongated objects (pen, fork, spoon, *etc*) and small objects (diameter $< 3\text{cm}$, *e.g.*, bottle cap). (3) **Viewpoint Challenge:** While keeping the coordinate-system origin fixed, we rotate the camera viewpoint around the $z$-axis (perpendicular to the tabletop) by $5°$, $15°$, and $30°$, respectively. (4) **Appearance-Deprived Challenge:** Scenes designed to weaken informative 2D cues, including *transparent objects*, *textureless objects* (solid white, solid green, solid blue, *etc*), and *visual camouflage* (objects with the same color as the tabletop), forcing the model to rely more on 3D geometry rather than 2D color and texture information. The real-world evaluation includes **47 distinct objects** (including containers). Each subtask is repeated twice, totaling 120 trials; each trial allows up to three grasping attempts.

We compare ANY3D-VLA with multiple strong 2D/3D VLAs, with results summarized in Figure 2. ANY3D-VLA outperforms all baselines across four real-world evaluation scenarios. In particular, the overall average success rate for (Setting 2, DA3) reaches 62.5%, representing a 29.2% improvement over the strongest baseline SpatialVLA, which achieves 33.3%. When the point-cloud source at inference is held fixed, hybrid point cloud training (Setting 2) typically achieves higher average success rates than training with simulator-only point clouds (Setting 1). Meanwhile, under the same training setting, inference with model-estimated point clouds (DA3) often outperforms inference with RealSense point clouds, consistent with the observation that state-of-the-art depth/point-cloud estimation models can often produce more accurate point clouds than commodity depth sensors. This also supports our training strategy: hybrid point cloud training effectively mitigates domain gaps induced by environmental differences and depth biases, while reducing reliance on depth sensors at deployment time. We also conduct a qualitative analysis to highlight the robustness of our method compared to baselines and to discuss shared limitations (Appendix J).

### 6.1.3. REAL-WORLD POST-TRAINING

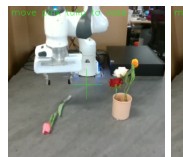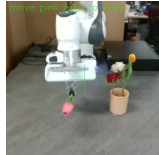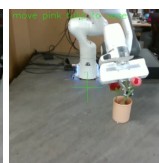

*Figure 3.* Example of Task 1: "Move pink tulip to vase".

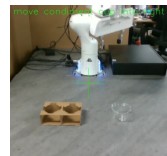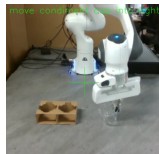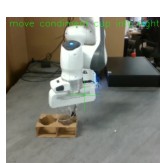

*Figure 4.* Example of Task 2: "Move condiment cup into right slot of cup carrier".

To adapt to more diverse real-world tasks, we employ a two-stage training paradigm: imitation learning pre-training on large-scale synthetic data, followed by fine-tuning on a limited amount of real-world data. To validate the model's generalization capabilities on new tasks including specific rules and new language instructions, we design two challenging evaluation scenarios: (1) grasping a flower and placing it into a vase (Figure 3), and (2) placing a transparent condiment cup into a specific slot of a cup carrier (Figure 4). For each task, we collect 100 demonstrations, with object positions and orientations randomly sampled within defined ranges. Figure 6 (Appendix B) shows examples from our real-world post-training dataset. During the

post-training phase, all models are trained exclusively on trajectories (whereas in the pre-training phase, both GraspVLA and our model utilize joint training with bounding boxes and trajectories). We train ANY3D-VLA under two data settings: pure RealSense or hybrid point clouds (30% RealSense, 20% UniDepthV2, 20% Depth Anything 3, 20% MapAnything). We then evaluate and compare its performance when using raw RealSense point clouds versus point clouds estimated by Depth Anything 3. During evaluation, we conduct 15 trials for each task, with three grasp attempts allowed per trial.

The evaluation results are reported in Table 3. ANY3D-VLA outperforms the baselines on both tasks. Under the same point-cloud source at inference time, hybrid point cloud training consistently performs better than training with RealSense point clouds only, achieving the best results when using DA3 for inference (Task1:93.3%/Task2:86.7%). Moreover, regardless of the training setting, DA3-based inference is generally no worse than, and often better than RealSense-based inference, indicating that hybrid point cloud training effectively mitigates the domain gap and reduces reliance on depth sensors during deployment.

*Table 3.* Success rates of post-training tasks. During inference, *RealSense* refers to the sensor-based point cloud, while *DA3* refers to the point cloud derived from Depth Anything 3 depth predictions.

| Model | Task 1 SR (%) | Task 2 SR (%) |
|---|---|---|
| $\pi_{0.5}$ | 33.3 | 26.7 |
| GraspVLA | 33.3 | 53.3 |
| SpatialVLA | 13.3 | 6.7 |
| ANY3D-VLA *(Setting 3, trained with RealSense point clouds)* | | |
| *RealSense* | 73.3 | 60.0 |
| *DA3* | 80.0 | 60.0 |
| ANY3D-VLA *(Setting 2, trained with hybrid point clouds)* | | |
| *RealSense* | 80.0 | 66.7 |
| *DA3* | **93.3** | **86.7** |

### 6.2. Inference Efficiency and Latency

To evaluate the computational overhead of introducing point-cloud representations, we treat inference latency as a metric. Table 4 presents the efficiency-performance trade-off across different visual input sources in the real world.

*Table 4.* Efficiency-performance trade-off. Inference speeds are measured on a single NVIDIA RTX 3090 GPU.

| Depth Source | Inference Speed | Point Cloud Size | Remarks |
|---|---|---|---|
| 2D baseline (GraspVLA) | 3.0 FPS | – | Fastest |
| RealSense | 2.0 FPS | ~3k–8k points | Faster |
| Depth Anything 3 | 1.7 FPS | ~3k–8k points | Better accuracy–latency trade-off |
| UniDepthV2 | 0.5 FPS | ~3k–8k points | High latency |
| MapAnything | 0.3 FPS | ~3k–8k points | Highest latency |

The raw point cloud is first processed through cropping

and 3D compression, significantly reducing it from approximately 30k–60k points to about 3k–8k points. In actual real-world deployment, we ultimately select Depth Anything 3 because it provides a balance between geometric accuracy and latency.

While estimating point clouds inherently introduces computational overhead compared to purely 2D approaches, we employ action chunking with a chunk size of 4, which effectively amortizes the perception latency. Furthermore, compared to higher-frequency policies, our model executes a larger motion per step (roughly 2–3× longer). As a result, an operating frequency of 1.7–2.0 FPS remains highly feasible for our target scenario of tabletop manipulation.

### 6.3. Diverse Point-Cloud Inputs as Data Augmentation

We conduct a quantitative analysis of ANY3D-VLA under different point cloud data compositions (simulator-based, sensor-based, model-estimated, and their mixtures). The results in Table 5 show that, whether evaluated in simulation (using simulator-based or model-estimated point clouds) or in real-world settings (using sensor-based or model-estimated point clouds), models trained with hybrid point clouds consistently outperform or match those trained with a single source under comparable conditions. These results suggest that exposing the model to diverse point-cloud inputs serves as an effective form of data augmentation, helping **mitigate the sim-to-real gap** and improve robustness. More broadly, this strategy not only **lessens reliance on depth sensors** in real-world deployment, but also strengthens generalization, enabling the model to **benefit from** higher-quality depth and point-cloud estimates produced by **future, more capable vision models**.

*Table 5.* Success rates of ANY3D-VLA across three different training configurations (*simulator only*, *hybrid point cloud*, *sensor only*). *Simulator*, *DA3*, *RealSense* denote three different point cloud sources used during inference.

|  | Simulator Only | Hybrid Point Cloud | Sensor Only |
|---|---|---|---|
| Simulation *(Test SR, %)* |  |  |  |
| *Simulator* | 80.0 | 81.1 | N/A |
| *DA3* | 78.9 | **82.1** | N/A |
| Real-World *(Zero-Shot)* *(Average SR, %)* |  |  |  |
| *RealSense* | 55.0 | 57.5 | N/A |
| *DA3* | 60.0 | **62.5** | N/A |
| Real-World *(Post-Training)* *(Task 1 SR, %)* |  |  |  |
| *RealSense* | N/A | 80.0 | 73.3 |
| *DA3* | N/A | **93.3** | 80.0 |

### 6.4. Ablation Study

To verify the necessity of our 2D–3D fusion design, we conduct an ablation study on the key components of the visual encoder under a setting where the model is trained purely on the simulator ground-truth point cloud. Table 6

reports the results on our simulation benchmark.

*Table 6.* Ablation study on the effect of 2D–3D fusion.

| Method | Single-Trial SR (%) | Test SR (%) | Grasp SR (%) |
|---|---|---|---|
| 2D-only (DINOv2-L+SigLIP) | 45.3 | 72.6 | 80.0 |
| 3D-only | 44.2 | 64.2 | **91.6** |
| 3D + SigLIP | 42.1 | 69.5 | 81.1 |
| 3D + (DINOv2-S+SigLIP) | 45.3 | 72.6 | 84.2 |
| Full 2D–3D fusion (3D + DINOv2-L+SigLIP) | **61.1** | **80.0** | 89.5 |

By comparing success rates and analyzing simulation test videos, we observe that the 3D-only model has relatively weak semantic understanding: it struggles to consistently localize the target object specified by the language instruction, and in some cases it grasps a different object yet still completes the subsequent procedure. Nevertheless, its Grasp SR is high (91.6%), indicating strong capability in fine-grained manipulation. In contrast, the 2D-only model is less stable in scenarios involving small objects or heavy occlusion. After introducing full 2D–3D fusion, the Single-Trial SR improves from 45.3% (2D-only) and 44.2% (3D-only) to 61.1%, with the other two metrics improving accordingly. These results suggest that: (1) relying solely on 3D representations is insufficient for robust manipulation; and (2) appropriately fusing 2D semantics with 3D geometry is critical for achieving robust manipulation.

### 6.5. LIBERO and CALVIN Benchmarks

We evaluate on two public simulation benchmarks: LIBERO (Object, Goal, Long, and Spatial) (Liu et al., 2023) and CALVIN (ABC→D) (Mees et al., 2022). Specifically, $\pi_{0.5}$ and SpatialVLA are fine-tuned from their publicly released pretrained weights, whereas GraspVLA and our model are first pretrained on our synthetic RGBD manipulation dataset and then fine-tuned. Similar to the real-world experiments, we uniformly set the action chunk size to 4, use the front-view as visual input, and freeze the 2D vision transformer. ANY3D-VLA achieves good results: it improves over GraspVLA by 13.9% on LIBERO; on CALVIN, it increases the average length by 0.71 compared to GraspVLA; and it is consistently slightly better than SpatialVLA overall. Details are provided in Appendix K.

## 7. Limitations and Future Work

Although we have evaluated this work in both simulation and real-world manipulation settings, several limitations remain: (1) Our real-world experiments currently cover only a single robotic arm and a limited set of objects. Future work could extend to additional robot platforms and environments, and evaluate more complex, long-horizon tasks. (2) We adopt a relatively simple 3D grid compression and

representation extraction scheme; stronger 3D backbones may yield additional gains. We intentionally avoid overly complex architectures to preserve the interpretability of our controlled comparisons on how observation spaces and representation design affect performance. (3) There exist many strategies for leveraging depth and point clouds, and we do not exhaust all possibilities. Our conclusion should be interpreted as follows: introducing mature, native sparse 3D point (compressed point clouds) representations is a promising direction, but the specific implementation proposed in this paper is **not the only viable choice**. More discussion of limitations and future work is provided in Appendix L.

## 8. Conclusion

This paper investigates the design of observation spaces and visual representations for VLA manipulation. While recent implicit and reconstruction-based spatial priors injected by models such as VGGT improve visual representations to some extent, our comparative experiments reveal clear limitations. In contrast, fusing native sparse 3D structure (*i.e.*, compressed point clouds) with the corresponding 2D patch representations provides more direct and stable spatial constraints and interaction geometry required for manipulation. To address the data-scaling bottleneck introduced by reliance on point clouds, we propose a hybrid point cloud training strategy and show that the model remains stable even when inference uses noisy point clouds. Experiments in both simulation and real-world demonstrate that ANY3D-VLA outperforms existing baselines in complex scenarios. Overall, we reaffirm that point-cloud representations matter: integrating mature point-cloud representations with 2D representations is an effective path to stronger spatial perception and manipulation in VLA models. Crucially, this approach does not require expensive depth hardware at deployment or stringent data collection. With hybrid point clouds, we can achieve more robust 3D geometric capability without significantly increasing the barriers to training or inference, laying the groundwork for general-purpose real-world manipulation.

## Acknowledgements

This work is supported by the National Natural Science Foundation of China (No. 62422606) and the Hong Kong Research Grant Council General Research Fund (No. 17213925).

## Impact Statement

This work aims to improve the robustness of robotic manipulation under challenging visual conditions (*e.g.*, noisy depth measurements and viewpoint variations), thereby enhancing safety and reliability for deployment in industrial, domestic, and assistive-care settings. More capable robots may affect certain manual-labor jobs and introduce new safety risks, such as unintended interactions with humans or fragile objects. Our experiments are conducted in controlled environments; for real-world deployment, we recommend incorporating appropriate safety monitoring, human oversight, and risk assessment.

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

## A. Extended Related Work

**2D–3D Representation Fusion.** The idea of fusing 2D image features and 3D geometric representations has been widely studied beyond VLA manipulation. ODIN (Jain et al., 2024) proposes a unified transformer for 2D and 3D segmentation by alternating between 2D within-view fusion and 3D cross-view fusion. BPNet (Hu et al., 2021) performs joint 2D–3D scene understanding through bidirectional projection modules that enable information exchange between 2D image and 3D point-cloud streams. RGB-D fusion has also been explored for point-cloud-based 3D human pose estimation by injecting color features extracted from RGB images into point-cloud representations (Ying & Zhao, 2021). In addition, the Concerto encoder used in our method is itself pretrained through joint 2D–3D self-supervised learning (Zhang et al., 2025; 2026). Therefore, our work does not claim the first conceptual use of 2D–3D fusion. Instead, we focus on how to instantiate this general fusion principle for VLA manipulation: we lift depth into native 3D space, compress the point cloud for efficient encoding, extract features using a pretrained 3D backbone, align the resulting 3D features back to 2D patches, and train under heterogeneous point-cloud sources to improve robustness during real-world deployment.

**Simulator, Sensor-Based, and Model-Estimated Depth for Point-Cloud Construction.** Recent work further suggests that language can serve as an additional prior for monocular depth estimation, especially for resolving metric-scale ambiguity and improving depth prediction in semantically described regions (Zeng et al., 2024b;c; Zhang & Lu, 2025; Zeng et al., 2024a; Cui et al., 2025). In this work, we use off-the-shelf depth estimators without conditioning them on task instructions, and instead rely on hybrid point-cloud training to improve robustness to their errors. Incorporating instruction- or scene-description-conditioned depth estimation into VLA deployment is an interesting future direction.

## B. Dataset and Benchmark Details

Figures 5 and 6 present example samples from our large-scale synthetic pre-training RGBD dataset and our real-world post-training RGBD dataset. Both datasets include depth from multiple sources, which we use to construct diverse point-cloud inputs.

## C. Pilot Study Details

We compare five different approaches. Each constructs the observation $o_t$ through a distinct perception and processing pipeline, differing in the degree of explicit geometric information and the form of representation. The detailed configurations are described below:

- **2D-only.** The observation consists only of RGB images $o_t^{\text{rgb}} = \{I_t^{(v)}\}_v$, where $I_t^{(v)}$ is the image captured at time $t$ from the $v$-th viewpoint. The model extracts features using a standard image encoder, without providing any depth or 3D geometric information.

- **Implicit-depth RGB (depth-pretrained image encoder).** The observation remains pure RGB images $o_t^{\text{impl-depth}} = \{I_t^{(v)}\}_v$, but we introduce two parallel branches: one branch is a conventional image encoder (DINOv2 + SigLIP), and the other is an encoder pre-trained on metric depth estimation (the encoder of Depth Anything v2). Both branches take the same set of RGB images as input. We discard the DPT (Ranftl et al., 2021) decoder of the latter and keep only its RGB feature extractor. The features from the two branches are concatenated along the patch level and then fed into the subsequent VLA backbone. Geometric information is thus injected as an *implicit depth prior*: while the input remains 2D, the visual representation carries a stronger depth bias.

- **Implicit-3D RGB (VGGT-style image encoder).** The observation is RGB images $o_t^{\text{impl-3d}} = \{I_t^{(v)}\}_v$, but we adopt a dual-branch structure: one branch is a standard image encoder, and the other is the encoder of a spatial foundation model trained with multi-view geometric reconstruction objectives (*e.g.*, predicting depth and camera poses), such as VGGT. The two branches share the same RGB inputs. We discard VGGT's geometry-prediction heads and keep only its image tokens, which are concatenated with the visual tokens from the standard branch at the patch level and fed into the VLA backbone. This observation space still contains only 2D visual tokens in form, but these tokens *implicitly* encode the 3D geometric priors learned by VGGT.

- **RGBD image-plane (explicit depth input + RGB).** The observation augments RGB images with depth channels $o_t^{\text{rgbd}} = \{[I_t^{(v)}, D_t^{(v)}]\}_v$, where $D_t^{(v)}$ denotes the depth map from the $v$-th view. We feed the depth maps into the **same**

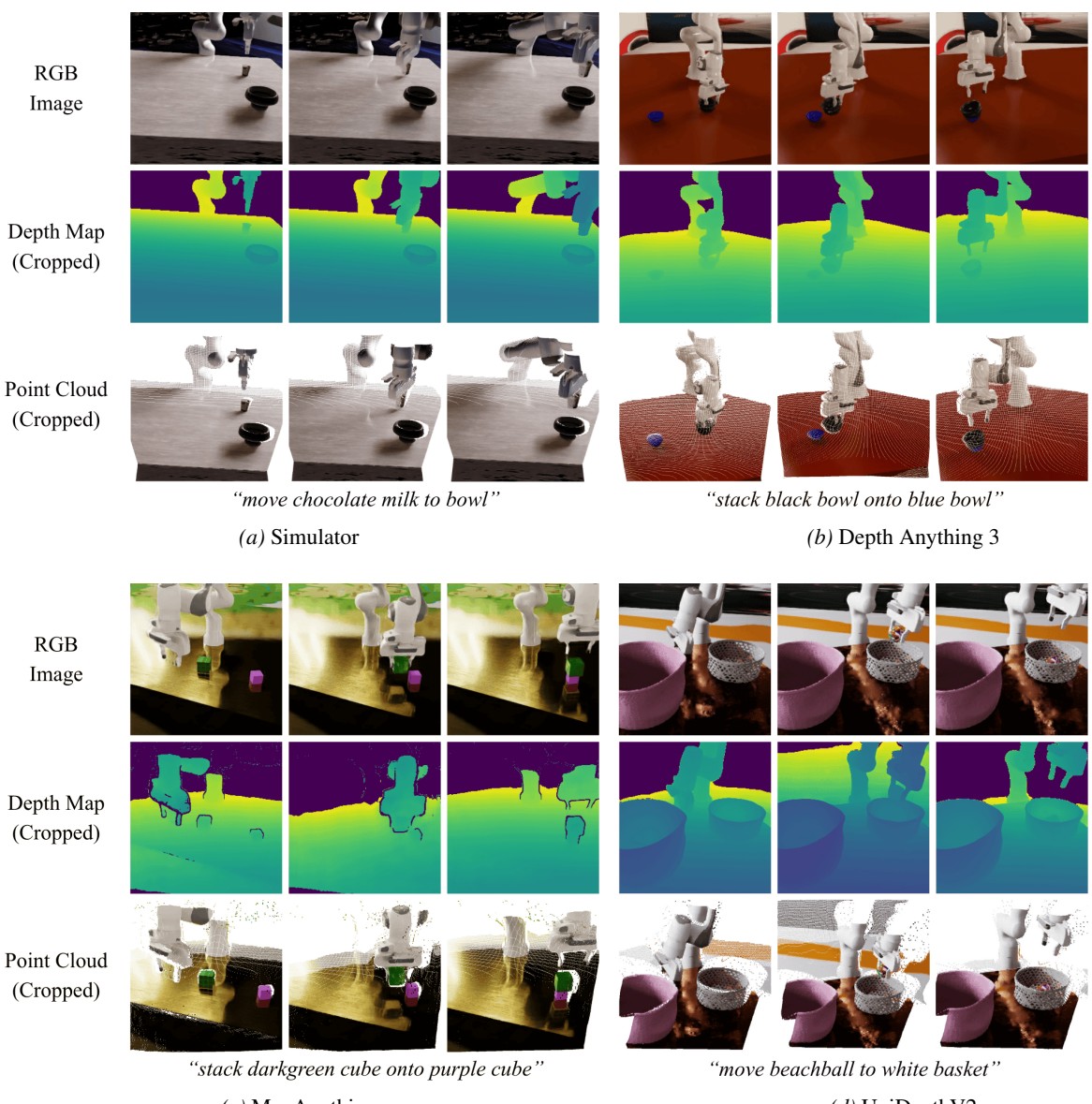

RGB
Image

Depth Map
(Cropped)

Point Cloud
(Cropped)

*"move chocolate milk to bowl"*

*(a)* Simulator

*"stack black bowl onto blue bowl"*

*(b)* Depth Anything 3

RGB
Image

Depth Map
(Cropped)

Point Cloud
(Cropped)

*"stack darkgreen cube onto purple cube"*

*(c)* MapAnything

*"move beachball to white basket"*

*(d)* UniDepthV2

*Figure 5.* Large-scale synthetic pre-training RGBD dataset. (a) shows the RGB images rendered by the simulator and the corresponding ground-truth metric depth maps, which are converted into point clouds using fixed camera intrinsics. (b)(c)(d) respectively present the metric depth maps estimated from a single RGB frame by Depth Anything 3, MapAnything, and UniDepthV2, along with the point clouds computed under the same camera intrinsics. To eliminate interference from background textures, we crop the depth maps in the dataset; therefore, the model inputs are RGB images and the cropped point clouds.

image encoder as RGB to extract depth tokens in the image plane, and then fuse them with RGB tokens at the patch level. Geometric information thus enters the VLA as an *image-plane RGBD representation*, rather than as native 3D structures (point clouds), forming a class of explicit depth VLAs.

- **Point cloud–2D patch fusion.** The observation $o_t^{\text{pc}}$ first lifts the RGBD inputs of each view into a point-cloud set $\mathcal{P}_t = \{(x_i, y_i, z_i, c_i, n_i)\}_i$ in the camera coordinate system using the camera intrinsics, where $(x_i, y_i, z_i)$ are 3D coordinates, $c_i$ is color, and $n_i$ denotes normals and other attributes. We then perform compression within the 3D space and extract 3D representations via a pre-trained point cloud encoder, which is further aligned and fused with 2D image representations at the patch level. This setting explicitly constructs and leverages 3D geometric structures.

RGB
Image

Depth Map
(Cropped)

Point Cloud
(Cropped)

*(a)* RealSense                                    *(b)* Depth Anything 3

RGB
Image

Depth Map
(Cropped)

Point Cloud
(Cropped)

*(c)* MapAnything                                   *(d)* UniDepthV2

*Figure 6.* Real-world post-training RGBD dataset example: "Move condiment cup into right slot of cup carrier." (a) The RGB image captured by the RealSense camera and its corresponding metric depth map, with the depth converted into a point cloud using fixed camera intrinsics. (b)(c)(d) show the metric depth maps estimated from a single RGB frame by Depth Anything 3, MapAnything, and UniDepthV2, respectively, along with the corresponding point clouds computed using the same camera intrinsics. To eliminate background interference, we crop the depth maps in the dataset; therefore, the model input consists of the RGB image and the cropped point cloud. As can be seen, compared with the model-estimated point clouds, the point clouds reconstructed by RealSense are coarser, which is particularly evident in regions containing transparent objects.

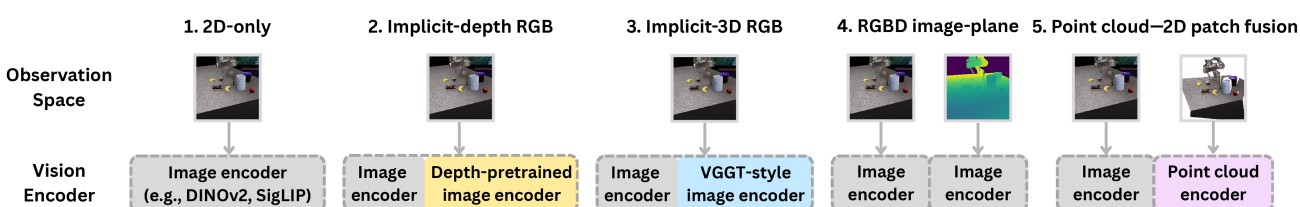

*Figure 7.* Different observation spaces and visual representations.

## D. Point-Cloud Construction and 3D Compression Details

We construct a point cloud and per-point attributes from a single-view RGBD observation. We first use a depth-validity mask to select all valid pixels $(u_{\text{valid}}, v_{\text{valid}})$ and their metric depths $z_{\text{valid}}$, and retrieve the corresponding colors from the RGB image using the same indices. We then apply a synchronized cropping operation $\texttt{keep}(\cdot)$ (based on the $y$-range, the radius in the $x$–$z$ plane, and the $z$-range) to filter out background while maintaining one-to-one alignment between geometry and color. Using the pinhole intrinsics $(f_x, f_y, c_x, c_y)$, we back-project the valid pixels into 3D to obtain $\texttt{coord} = (x_{\text{cam}}, y_{\text{cam}}, z)$ in the camera coordinate frame. The point color $\texttt{color}$ is obtained via the same pixel indices, forming aligned per-point attributes $[x, y, z, r, g, b]$. Next, we estimate a normal vector ($\texttt{normal}$) for each point by fitting a local neighborhood plane on the cropped dense point set, and use it as an additional per-point attribute for subsequent transforms. To obtain a more compact representation with more uniform spatial coverage, we voxelize points with voxel size $g = 1\text{cm}$ via $\mathbf{v}_i = \lfloor \mathbf{p}_i / g \rfloor$, and keep only one representative point in each non-empty voxel while inheriting its attributes such as color and normal (Wu et al., 2025). Finally, we record the inverse index from the original points to voxel cells as well as the discrete voxel coordinates, to support subsequent feature alignment and backfilling. Due to differences in depth sources and environmental variations (including simulation and real world), the cropped point cloud typically contains 30,000–60,000 points, while after 3D compression, it contains about 3,000–8,000 points.

## E. Vision Encoder Details

In this section, we provide the implementation details of the encoders used to construct the visual representations. The specific model choices are engineering configurations rather than conceptual contributions, and they can be replaced with more advanced pre-trained image and point-cloud encoders.

**2D Image Encoder.** For RGB images, we employ two pre-trained vision transformer models: DINOv2[1] (Oquab et al., 2024) and SigLIP[2] (Zhai et al., 2023). Both models are kept frozen during inference to serve as static feature extractors. We extract their intermediate layer feature maps from before the penultimate layer and remove the [CLS] token. Given images, we extract features through both models separately and then concatenate them along the feature dimension, with a total dimension of $D_{\text{dino}} + D_{\text{siglip}}$ (1024 + 1152 = 2176).

**3D Point Cloud Encoder.** For 3D input, we use a pre-trained point cloud encoder Concerto (Zhang et al., 2025). To balance computational efficiency and representation capability, the majority of the encoder's parameters are frozen, but the last 4 sparse convolution (spconv) layers are unfrozen for partial fine-tuning. The point cloud is first grid-sampled; the encoder then takes point coordinates, colors, and normals as input and produces a 3D feature $\mathbf{f}_i^{\text{3D}}$ (with dimension 1728) for each input point. We also define a learnable empty token, which represents 2D patches that do not have corresponding 3D points.

## F. ANY3D-VLA Training Details

### F.1. Joint Training of VLM and Action Expert

The model is jointly trained on the web-scale grounding dataset GRIT (Peng et al., 2023) and a synthetic VLA dataset. In each batch, samples from the two sources are randomly mixed: samples from GRIT are used only to supervise the bounding box prediction of VLM, whereas samples from the synthetic VLA dataset supervise bounding box tokens, grasp pose tokens, and the flow-matching objective of the continuous action expert. Let $\mathbf{h}_{\text{fused}}$ denote the sequence of fused visual tokens obtained after 2D–3D fusion, and let $x_{\text{text}}$ denote the sequence of input text instruction tokens.

**VLM loss.** The VLM autoregressively generates **discrete** bounding box tokens and grasp pose tokens. Let $N_{\text{bbox}}$ and $N_{\text{gpose}}$ denote the lengths of the bounding box and grasp pose sequences, and let $y_{\text{bbox},n}$ and $y_{\text{gpose},n}$ be the tokens at position $n$ in each sequence. The sequence loss $\mathcal{L}_{\text{S2}}$ is defined as

$$\mathcal{L}_{\text{S2}} = -\sum_{n=1}^{N_{\text{bbox}}} \log P_\theta(y_{\text{bbox},n} \mid \mathbf{h}_{\text{fused}}, x_{\text{text}}, y_{\text{bbox},<n}) - \mathbb{I}_{\text{synthetic}} \sum_{n=1}^{N_{\text{gpose}}} \log P_\theta(y_{\text{gpose},n} \mid \mathbf{h}_{\text{fused}}, x_{\text{text}}, y_{\text{bbox}}, y_{\text{gpose},<n}). \quad (4)$$

---

[1]vit_large_patch14_reg4_dinov2.lvd142m
[2]vit_so400m_patch14_siglip_224

Here, $\mathbb{I}_{\text{synthetic}}$ is an indicator function that equals $1$ if the sample is drawn from the synthetic dataset and $0$ otherwise. Consequently, for GRIT samples only the bounding box term is optimized, while for synthetic samples both the bounding box and grasp pose terms are optimized.

**Flow-matching loss for the action expert.** The continuous action head ("action expert") predicts chunked end-effector delta actions via flow matching. Let $\mathbf{A}_0 \in \mathbb{R}^{H \times d}$ denote the ground-truth action chunk of horizon $H$, where $d$ is the action dimension per step (*e.g.*, end-effector pose deltas and gripper command), and let $\mathbf{A}_t$ be its noised version at time $t \in [0, 1]$. The model-predicted vector field is denoted by $v_t(\mathbf{A}_t, \mathbf{h}_{\text{fused}}, y_{\text{bbox}}, y_{\text{gpose}})$, while $u_t(\mathbf{A}_t, \mathbf{A}_0)$ denotes the ground-truth vector field constructed by the flow. The flow-matching loss $\mathcal{L}_{\text{S1}}$ is defined as

$$\mathcal{L}_{\text{S1}} = \mathbb{I}_{\text{synthetic}} \, \mathbb{E}_{t \sim \mathcal{U}(0,1)} \left\| v_t(\mathbf{A}_t, \mathbf{h}_{\text{fused}}, y_{\text{bbox}}, y_{\text{gpose}}) - u_t(\mathbf{A}_t, \mathbf{A}_0) \right\|_F^2. \tag{5}$$

In practice, for each sample, we draw a single time step $t$ from a uniform distribution over $[0, 1]$ to approximate the above expectation.

**Total loss.** The overall training objective is the sum of the sequence loss and the flow-matching loss:

$$\mathcal{L}_{\text{total}} = \mathcal{L}_{\text{S2}} + \mathcal{L}_{\text{S1}}. \tag{6}$$

Notably, we avoid explicit supervision for depth maps or point clouds, allowing them to affect the model only through the fused features $\mathbf{h}_{\text{fused}}$. This ensures improvements are ascribed directly to better spatial observations and visual representations.

### F.2. Data Augmentation and Pose Perturbations

In the image preprocessing stage, we adopt random padding: we first resize the image while preserving its aspect ratio, then randomly translate it onto a fixed-size canvas to perturb the target locations. For the robot end-effector pose, we inject uniform noise into the translation and rotation of the trajectories during training to improve robustness to pose perturbations.

### F.3. Hybrid Point Cloud Training Details

The mixing ratios across different point-cloud sources are reported in Table 7. Figure 8 shows the noise and scale differences across various depths.

*Table 7.* Distribution and model specifications of the hybrid training dataset. The point-cloud mixing ratios are consistent between the pre-training and post-training datasets.

| Source | Ratio | Model ID |
|---|---|---|
| Simulator / Sensor | 30% | N/A |
| UniDepthV2 | 30% | `lpiccinelli/unidepth-v2-vitl14` |
| Depth Anything 3 | 20% | `depth-anything/da3metric-large` |
| MapAnything | 20% | `facebook/map-anything` |

### F.4. Optimization Details.

We use the AdamW optimizer (Kingma, 2014) with a learning rate of $1.6 \times 10^{-4}$, $\beta = (0.9, 0.999)$, and weight decay set to $0$. The final model is trained on 16 H800 GPUs for 3 days with a global batch size of 128.

## G. Evaluation of ANY3D-VLA in the Simulator with Point Clouds from Different Sources

As shown in Table 8, we evaluate ANY3D-VLA in the simulation environment by fixing the RGB input while utilizing point clouds generated by different models. The results demonstrate that ANY3D-VLA maintains competitive performance even with imperfectly estimated point clouds.

## H. Detailed Training Setting for ANY3D-VLA and Baseline Models

Detailed hyperparameters for ANY3D-VLA and all baseline models are provided in Table 9. We train $\pi_{0.5}$ (Black et al., 2025a) and SpatialVLA (Qu et al., 2025) using pre-trained weights for initialization. For all models, we consistently **freeze**

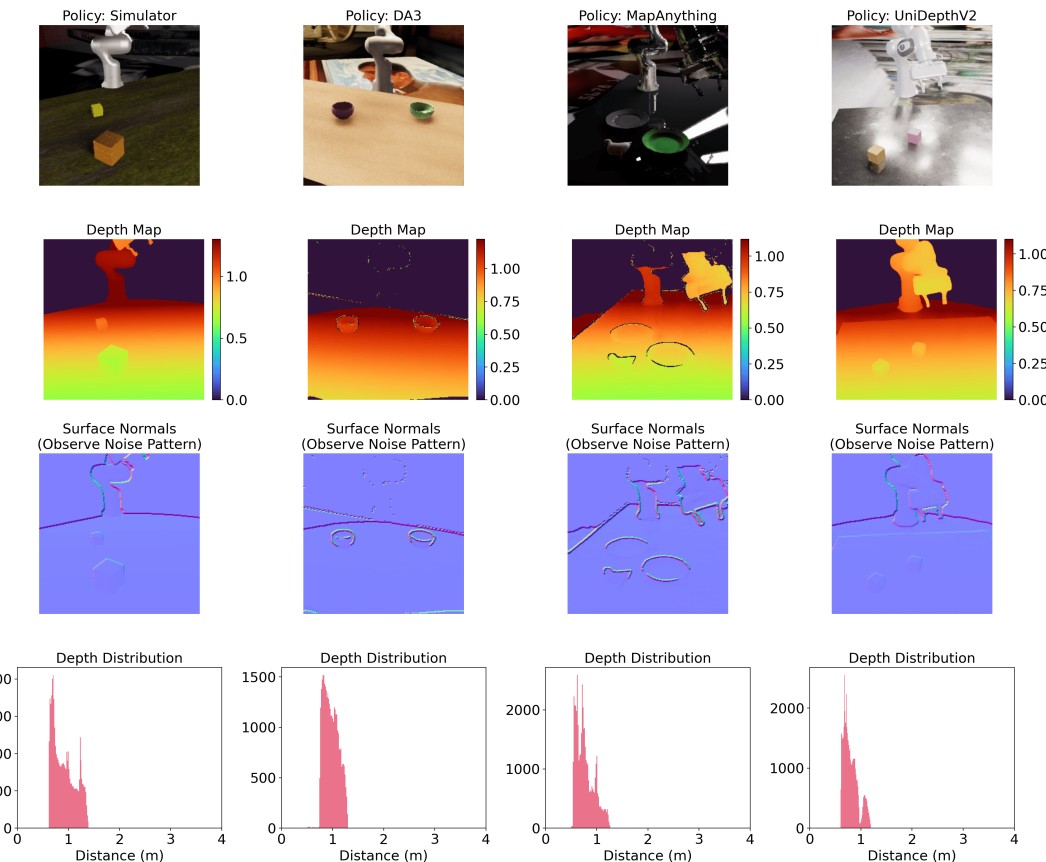

*Figure 8.* Sampling examples and multi-dimensional analysis of the pre-training simulation dataset. Our large-scale synthetic RGB-D dataset comprises four depth sources with distinct characteristics: Simulator, DA3 (Depth Anything V3), MapAnything, and UniDepthV2. The first two rows display the RGB images and their corresponding depth maps. The third row analyzes disparities in depth noise: surface normal maps reveal that, compared to the smoothness of the Simulator, model-generated depth maps introduce varying degrees of observation noise and edge artifacts, which helps simulate the non-ideal characteristics of real-world sensors. The fourth row illustrates differences in distribution and range: depth distribution histograms highlight significant variations in numerical scales across different sources. This diverse data composition enhances the VLA model's generalization capability across varying depth scales and point cloud inputs.

*Table 8.* Evaluation of ANY3D-VLA (trained purely on simulator point cloud) in the simulator using the same RGB input with point clouds from different sources. SR: success rate.

| Source During Evaluation | Single-Trial SR (%) | Test SR (%) | Grasp SR (%) | Inference Speed (FPS) |
|---|---|---|---|---|
| Simulator | 61.1 | 80.0 | **89.5** | **2** |
| UniDepthV2 | 62.1 | **81.1** | 87.4 | 0.5 |
| Depth Anything 3 | 61.1 | 78.9 | 87.4 | 1.7 |
| MapAnything | **63.2** | 80.0 | **89.5** | 0.3 |

**the 2D vision transformer** during both the pre-training and post-training stages. Since $\pi_{0.5}$ and SpatialVLA do not support a co-training mechanism, we do not jointly train them with the GRIT dataset. Instead, they are pre-trained only on our large-scale synthetic VLA dataset.

# I. Real-world Deployment Setup

We deploy our model on a single NVIDIA RTX 3090 GPU. With RealSense depth as input, ANY3D-VLA runs at 2 FPS, while with Depth Anything 3 depth as input, ANY3D-VLA runs at 1.7 FPS. All real-world experiments are conducted on

*Table 9.* Hyperparameters for different methods during both the pre-training and post-training stages.

| Method | Batch | LR | Schedule / Optimizer | Pre-trained Checkpoint |
|---|---|---|---|---|
| $\pi_{0.5}$ | 256 | Peak LR: 2.5e-5 Decay LR: 2.5e-6 | Cosine Schedule Warmup Steps: 1k Decay Steps: 30k | pi05_base |
| GraspVLA | 128 | 1.6e-4 | Constant | N/A |
| SpatialVLA | 128 | 2e-5 | Linear Schedule Warmup Ratio: 0.005 Weight Decay: 0.0 | IPEC-COMMUNITY/ spatialvla-4b-224-pt |
| ANY3D-VLA | 128 | 1.6e-4 | Constant | N/A |

a Franka Panda arm, with an external front-view Intel RealSense D435 camera providing monocular observations. The camera extrinsics are calibrated to align with the camera distribution used in the synthetic data. Unless otherwise specified, the camera is placed near the center of the synthetic sampling range. Test objects are placed within a $40\text{cm} \times 50\text{cm} \times 20\text{cm}$ workspace in front of the robot (Figure 2).

## J. Qualitative Analysis and Limitations

We analyze failure cases to compare baseline models ($\pi_{0.5}$, GraspVLA, SpatialVLA) with our method and to further clarify the limitations.

Across a variety of scenarios, the baseline models often exhibit horizontal grasp-position drift (*e.g.*, selecting a grasp point biased toward the front-view camera), indicating errors in spatial localization. This issue becomes particularly pronounced when the robot base is occluded by the target object or a container: occlusion further amplifies the grasp-point offset, suggesting that the models may rely on non-robust spatial shortcut cues. In appearance-deprived scenarios, where the target object is overly similar to the background in color and texture, the baselines are more prone to target localization failures, leading to unsuccessful grasps. In contrast, introducing point-cloud representations substantially mitigates these issues, highlighting the advantages of our method in spatial understanding and robustness.

Meanwhile, we also observe several failure modes shared across models. First, when grasping objects with smooth surfaces (*e.g.*, a plastic strawberry), the object can slip during lifting; incorporating tactile feedback or more fine-grained grip-force control may help alleviate this problem. Second, when the target is occluded, models struggle to grasp precisely; active perception (*e.g.*, changing viewpoints to reduce occlusion) or adding more cameras may provide a better solution. Third, in a small number of cases, the gripper collides with the environment and becomes stuck; addressing such issues may require reinforcement learning or planning-and-control strategies with explicit collision constraints.

## K. Comparisons on LIBERO and CALVIN Benchmarks

We evaluate on two public simulation benchmarks: LIBERO (diverse generalization capabilities) (Liu et al., 2023) and CALVIN (long-horizon task execution) (Mees et al., 2022). Specifically, $\pi_{0.5}$ and SpatialVLA are fine-tuned from their publicly released pretrained weights, whereas GraspVLA and our model are first pretrained on our synthetic RGBD manipulation dataset and then fine-tuned (because GraspVLA's released pretrained weights are tailored only to a dual-view setup and grasping tasks). Similar to the real-world experiments, we uniformly set the **action chunk** size to **4**, use the **front-view** as visual input, and **freeze** the 2D vision transformer. We construct point-cloud inputs using the simulator-provided metric depth and camera intrinsics.

**LIBERO.** We evaluate on four LIBERO suites (Object, Goal, Long, and Spatial). During fine-tuning, we jointly train on tasks from all four suites. Each LIBERO task is evaluated under 50 random initial states. For each initial state, we run one full episode and determine whether the task is completed within a maximum step budget. We summarize results using success rate (number of successful episodes / total episodes), and report the average across all tasks and initial states. As shown in Table 10, ANY3D-VLA achieves good performance: it outperforms GraspVLA by 13.9%, and its best average success rate is slightly higher than SpatialVLA's (by 0.3%).

*Table 10.* Comparisons on the LIBERO benchmark. During inference, *DA3* refers to the point cloud derived from Depth Anything 3 depth predictions, while *Simulator* refers to the simulator-based point cloud.

| Method | Success Rate (%) | | | | |
|---|---|---|---|---|---|
| | Object | Goal | Long | Spatial | Avg. |
| Maximum Step | 280 | 300 | 520 | 220 | 330 |
| $\pi_{0.5}$ | **80.6** | **92.6** | **81.2** | 77.0 | **82.9** |
| GraspVLA | 58.8 | 70.4 | 34.8 | 54.4 | 54.6 |
| SpatialVLA *(full-tuning)* | 73.2 | 78.2 | 43.8 | **77.4** | 68.2 |
| ANY3D-VLA *(trained with simulator point clouds)* | | | | | |
|   *DA3* | 76.4 | 80.8 | 39.8 | 74.0 | 67.8 |
|   *Simulator* | 77.2 | 80.2 | 40.8 | 75.6 | 68.5 |

**CALVIN.** CALVIN comprises 34 distinct tasks, covering a diverse range of skills from basic pick-and-place operations to the manipulation of articulated objects. The benchmark includes four different environments, each equipped with a Franka Panda arm for tabletop manipulation tasks. We adopt a challenging evaluation setup: the policy is trained using demonstration data from environments A, B, and C, and then subjected to a zero-shot evaluation in environment D. This evaluation utilizes a test set of 1,000 unique instruction chains, each composed of five sequential tasks, to rigorously measure the policy's generalization capabilities. As shown in Table 11, ANY3D-VLA achieves good results, improving the best average length by 0.71 compared to GraspVLA.

*Table 11.* Comparisons on the CALVIN benchmark.

| Method | Tasks Completed in a Row | | | | | Avg. Len |
|---|---|---|---|---|---|---|
| | 1 | 2 | 3 | 4 | 5 | |
| $\pi_{0.5}$ | **86.7** | **73.2** | **61.8** | **54.3** | **45.9** | **3.22** |
| GraspVLA | 56.2 | 47.5 | 40.9 | 30.0 | 22.5 | 1.97 |
| SpatialVLA *(full-tuning)* | 70.0 | 62.5 | 54.5 | 48.0 | 32.2 | 2.67 |
| ANY3D-VLA *(trained with simulator point clouds)* | | | | | | |
|   *DA3* | 69.1 | 57.1 | 50.0 | 44.4 | 32.0 | 2.53 |
|   *Simulator* | 72.7 | 61.5 | 52.9 | 46.2 | 34.8 | 2.68 |

**Analysis of the Performance Gap to $\pi_{0.5}$.** ANY3D-VLA still lags behind $\pi_{0.5}$ on LIBERO and CALVIN, which may be attributed to the following factors. First, unlike our real-world setting (§6.1.1), we did not retrain $\pi_{0.5}$ on our large-scale synthetic RGBD dataset for the public benchmarks. $\pi_{0.5}$ is pretrained on a richer and more diverse corpus that covers a wider range of manipulation scenarios and more complex language instructions, making it easier to adapt to the task distributions of LIBERO and CALVIN. Second, the public benchmark environments differ substantially from our large-scale synthetic data in terms of action space, gripper parameters, and camera field of view, which constrains the performance of ANY3D-VLA. In contrast, in our real-world comparisons where the data mismatch is eliminated, we demonstrate the advantages of our observation space and representation design. Therefore, this performance gap primarily reflects a mismatch between the data and environment configurations, rather than a limitation of the proposed method itself.

To further demonstrate the generalizability of our approach, we introduced a 3D branch into the $\pi_{0.5}$ backbone. As shown in Table 12, incorporating the 3D branch yields performance improvements on both the LIBERO and CALVIN benchmarks. On LIBERO, the average success rate increases from 82.9% to 87.1% (using DA3 depth for inference) and 87.3% (using Simulator depth for inference). On CALVIN, the average completion length improves from 3.22 to 3.41 and 3.43, respectively. These results indicate that our 3D representations can consistently enhance the performance of strong 2D VLA baselines.

## L. Additional Discussion: Limitations and Future Work

An important future direction is to further disentangle localization from spatial understanding. Spatial foundation models such as VGGT provide strong correspondences and localization cues on the image plane, but they may not produce representations that are optimal for manipulation. Our results suggest that lifting depth into native 3D space and learning compressed embeddings in that space better matches the needs of manipulation.

*Table 12.* Performance improvements when introducing the 3D branch into the backbone $\pi_{0.5}$ on the LIBERO and CALVIN benchmarks.

| LIBERO Benchmark (Success Rate %) | | | | | |
| --- | --- | --- | --- | --- | --- |
| Method | Object | Goal | Long | Spatial | Avg. |
| $\pi_{0.5}$ | 80.6 | 92.6 | 81.2 | 77.0 | 82.9 |
| ANY3D-VLA ($\pi_{0.5}$ *backbone, trained with simulator point clouds*) | | | | | |
| *DA3* | 85.4 | 95.8 | 85.8 | 81.2 | 87.1 |
| *Simulator* | 85.8 | 95.8 | 86.2 | 81.2 | 87.3 |

| CALVIN Benchmark (Tasks Completed in a Row) | | | | | | |
| --- | --- | --- | --- | --- | --- | --- |
| Method | 1 | 2 | 3 | 4 | 5 | Avg. Len |
| $\pi_{0.5}$ | 86.7 | 73.2 | 61.8 | 54.3 | 45.9 | 3.22 |
| ANY3D-VLA ($\pi_{0.5}$ *backbone, trained with simulator point clouds*) | | | | | | |
| *DA3* | 89.4 | 77.0 | 66.1 | 58.8 | 49.6 | 3.41 |
| *Simulator* | 90.0 | 77.8 | 66.1 | 58.4 | 50.2 | 3.43 |

Hybrid point cloud training also raises a more general question: how should different observation sources be scheduled and weighted during pre-training and post-training to maximize downstream robustness? We show that a simple mixture of simulator-based (or sensor-based) and multiple model-estimated point clouds already brings benefits, and future work can explore more refined scheduling and weighting strategies.

Furthermore, several non-VLA architecture robot policies also utilize depth or point clouds for geometric reasoning (Ke et al., 2025; Ze et al., 2024; Qian et al., 2025; Gervet et al., 2023; Murali et al., 2025; Fang et al., 2020). Future work could explore applying the methods presented in this paper to such models to further unlock the potential of robotic manipulation.

