# OpenReview forum: "Any3D-VLA: Enhancing VLA Robustness via Diverse Point Clouds"
_ICML.cc/2026/Conference — ICML 2026 regular_

### Official Review · Reviewer_NMdp · 2026-03-05

**Soundness:** 3
**Presentation:** 4
**Significance:** 3
**Originality:** 3
**Overall Recommendation:** 4
**Confidence:** 4

**Summary:**

This paper mainly introduces a visual language action model called ANY3D-VLA, which improves robustness in various environments by integrating 2D and 3D point cloud information, especially when dealing with challenges such as small objects, perspective changes, and occlusion. Research has pointed out that traditional models have limitations in spatial understanding, while ANY3D-VLA enhances this ability by introducing diverse 3D point cloud sources (including simulators, sensors, and model estimation), and demonstrates its effectiveness and superiority in object grasping tasks through a series of experiments.

**Compliance With Llm Reviewing Policy:**

Affirmed.

**Final Justification:**

The rebuttal fully resolved my concerns. I maintain my original Weak Accept recommendation. The paper is technically solid, makes a meaningful contribution to integrating 3D reasoning into VLA models, and demonstrates promising real-world results.

**Key Questions For Authors:**

Please see the weaknesses.

**Limitations:**

yes

**Strengths And Weaknesses:**

Strengths:

1. The paper clearly identifies a major limitation of current VLA models: their reliance on 2D visual representations, which weakens spatial reasoning. The motivation for introducing explicit 3D structure is convincing and well-supported by the pilot study.

2. ANY3D-VLA is designed as a plug-in pipeline that can augment existing VLA backbones. Key design choices are elegant and intuitive.

3. The hybrid training strategy that mixes point clouds from multiple sources (simulator, sensor, model-estimated) is a practical solution to the sim-to-real gap in 3D data. This design is useful because real robotic systems often lack reliable depth sensors.

4. The paper provides experimental results under various settings: 1) Pilot study comparing different 3D representations; 2) Simulation; 3) Real-world zero-shot evaluation; 4) Few-shot real-world fine-tuning. The real-world experiments with viewpoint perturbations, scale variation, and appearance-deprived settings strengthen the empirical validity.

5. The writing of the paper is well organized and easy to follow.

Weaknesses:

1. Introducing a point cloud encoder and 3D compression pipeline likely increases memory usage and latency. However, the paper does not provide detailed runtime or efficiency comparisons, which are important for real robot deployment.
2. The paper could benefit from deeper ablations on: different 3D encoders, different fusion mechanisms, the effect of 3D compression resolution, and point cloud density. These analyses would help clarify which components contribute most.

Overall, the paper presents a meaningful step toward integrating 3D reasoning into VLA models and demonstrates promising real-world results. Therefore, I give a positive rating score.

---

> ### Author Rebuttal · Authors · 2026-03-27
>
> Thank you for your positive review and recognition of our paper. We appreciate your detailed feedback and will address your concerns as follows.
>
> W1. We have already reported the runtime in the Table 7, Appendix F. When deployed on a single RTX 3090, Any3D-VLA runs at 2 FPS with RealSense depth input, and at 1.7 FPS when depth is estimated by Depth Anything 3. We also compared the inference speed of different point-cloud sources: Simulator / DA3 / UniDepthV2 / MapAnything achieve 2.0 / 1.7 / 0.5 / 0.3 FPS, respectively. This is also why we ultimately chose DA3 in our real-world experiments: it provides the most suitable trade-off between point-cloud quality and inference latency.
>
> W2. We agree with the reviewer’s suggestion that further analysis of different 3D encoders, different fusion strategies, different compression resolutions, and different point-cloud densities would all be highly valuable directions for future work, and we will add this more explicitly. The goal of the current work, however, is to first answer a more fundamental question: for VLA manipulation, what kind of observation space / representation design is more effective? For this reason, we intentionally keep the 3D implementation relatively simple, so that the study remains interpretable and controlled, rather than attempting to exhaust the entire 3D design space at once. We have also explicitly noted in the limitations that stronger 3D backbones and other depth/point-cloud utilization strategies may yield further gains in the future. We hope the paper will be understood as showing that native sparse 3D point representations are a promising direction, while the specific implementation proposed here is not the only viable solution.

---

> > ### Author Rebuttal · Reviewer_NMdp · 2026-04-03
> >
> > Thank you for the rebuttal. The additional runtime details and clarification regarding the scope of ablations address my concerns. I will maintain my original score.

---

> > > ### Author Response · Authors · 2026-04-04
> > >
> > > We appreciate your recognition of our work and the effort you put into the rebuttal!

---

### Official Review · Reviewer_wj8W · 2026-03-10

**Soundness:** 2
**Presentation:** 3
**Significance:** 3
**Originality:** 2
**Overall Recommendation:** 4
**Confidence:** 3

**Summary:**

This paper presents ANY3D-VLA, a modular framework designed to enhance the spatial understanding and robustness of VLAs by integrating native 3D point cloud representations. The authors address the challenges of 3D data scarcity and domain gaps by proposing a hybrid point cloud training strategy that unifies simulator-based, sensor-based, and model-estimated depth sources. The architecture employs a gated residual fusion mechanism to align 3D point-wise features with 2D patch tokens, allowing the model to leverage geometric priors without discarding pre-trained 2D semantics. Experiments in simulation and real-world environments demonstrate that ANY3D-VLA significantly outperforms 2D-based baselines.

**Compliance With Llm Reviewing Policy:**

Affirmed.

**Final Justification:**

I appreciate the authors’ detailed rebuttal. My concerns regarding the baseline methods and backbone have been addressed. I will raise my score to Weak Accept.

**Key Questions For Authors:**

1. In the hybrid training strategy, you used fixed mixing ratios for different depth sources. Did you find the model performance to be sensitive to these specific ratios?
2. In cases where the 3D-only model achieved high Grasp SR but lower task success, what specific semantic failures were most common?

**Limitations:**

Yes.

**Strengths And Weaknesses:**

Strengths:

1. The introduction of a hybrid point cloud training strategy is reasonable.
2. Introducing 3D information into VLA is interesting and important.
3. The introduction of a hybrid point cloud training strategy is useful.

Weaknesses:

1. I have concerns regarding the choice of baseline methods. The proposed method aims to inject 3D information into VLA, a goal shared by previous works such as PointVLA. However, the current evaluation predominantly focuses on RGB-only models, with the exception of SpatialVLA. To better position the contribution, the authors should provide more direct comparisons with specialized 3D-injection methods such as BridgeVLA, SpatialForcing, etc [1][2].
2. The backbone selected for the framework  appears less competitive compared to state-of-the-art VLA models like $\pi_{0.5}$ or Wall-OSS. Given that the primary claim is the effectiveness of 3D information injection, utilizing a more modern and powerful base structure (e.g., the open-sourced $\pi_{0.5}$) would provide a more convincing demonstration of the method's scalability.
3. According to Table 9 and Table 10, the method's performance is inferior to $\pi_{0.5}$ on the CALVIN and LIBERO benchmarks. While the authors attribute this to mismatched data distributions, it raises questions about the generalizability of the proposed features. A more rigorous test would be to fine-tune $\pi_{0.5}$ using the proposed 3D-branch to see if it yields a net gain over the original SOTA model.
4. Given that the main contribution is the 3D information injection, the structural analysis of 3D embeddings is somewhat weak.

[1] Bridgevla: Input-output alignment for efficient 3d manipulation learning with vision-language models.

[2] Spatial forcing: Implicit spatial representation alignment for vision-language-action model.

---

> ### Author Rebuttal · Authors · 2026-03-28
>
> W1.
> We agree that recent 3D VLAs deserve explicit discussion. Related Work section already covers Spatial Forcing, PointVLA, 3DS-VLA, and other approaches using depth or point clouds. Our experiments also include SpatialVLA as a 3D baseline. We do not include all recent 3D methods in the main experiments, as many differ substantially in pretraining data, robot embodiment, and other settings, making strictly controlled comparisons difficult. We will further clarify distinctions from prior work: BridgeVLA focuses on input–output alignment, while Spatial Forcing emphasizes implicit geometric representation alignment. Our work centers on explicit fusion between native 3D and 2D patch representations, along with hybrid training.
>
> W2.
> We consider this a valid concern, but it is orthogonal to the main claim of this paper. Compared to $\pi_{0.5}$, the original pretraining dataset of GraspVLA is more similar to that of Any3D-VLA (primarily simulation-based), making it a more appropriate base for introducing a 3D branch. Since Any3D-VLA is designed as a plug-in pipeline for existing backbones, we aim to avoid confounding factors from backbone differences that could obscure the core conclusions. Even under this controlled setting, Any3D-VLA demonstrates clear advantages in the critical real-world evaluations. In zero-shot evaluation, the best configuration outperforms the strongest baseline by 29.2%. In post-training tasks, the best results reach 93.3% / 86.7%. These gains indicate that the improvements primarily stem from the 3D representation and training strategy, rather than reliance on specific backbone choices. Additionally, the publicly available pre-trained version of $\pi_{0.5}$ uses a default setting with three input views, under which introducing a 3D branch makes it harder to clearly demonstrate the benefits of our method. Extending our 3D branch to stronger VLAs is an important direction for future work, and we will explicitly clarify this in the revised version.
>
> W3.
> GraspVLA is structurally equivalent to Any3D-VLA with the 3D branch removed. The two models share identical training pipelines and datasets, ensuring a fair comparison. Experimental results show that Any3D-VLA outperforms GraspVLA on LIBERO and CALVIN. In addition, introducing a 3D branch into the SOTA backbone $\pi_{0.5}$ also yields significant performance improvements, as shown below:
>
> Libero
> | Method | Object | Goal | Long | Spatial | Avg. |
> |---|---:|---:|---:|---:|---:|
> | $\pi_{0.5}$ | 80.6 | 92.6 | 81.2 | 77.0 | 82.9 |
> | $\pi_{0.5}$ + DA3 | 85.4 | 95.8 | 85.8 | 81.2 | 87.1 |
> | $\pi_{0.5}$ + Simulator | 85.8 | 95.8 | 86.2 | 81.2 | 87.3 |
>
> Calvin
> | Method | 1 | 2 | 3 | 4 | 5 | Avg. Len |
> |---|---:|---:|---:|---:|---:|---:|
> | $\pi_{0.5}$ | 86.7 | 73.2 | 61.8 | 54.3 | 45.9 | 3.22 |
> | $\pi_{0.5}$ + DA3 | 89.4 | 77.0 | 66.1 | 58.8 | 49.6 | 3.41 |
> | $\pi_{0.5}$ + Simulator | 90.0 | 77.8 | 66.1 | 58.4 | 50.2 | 3.43 |
>
> W4.
> The current paper already provides structural observation. In the ablation study, 3D-only model achieves only 44.2% Single-Trial SR, but reaches 91.6% Grasp SR. Its main issue lies in insufficient semantic understanding: it may sometimes grasp the wrong object, while still executing stable manipulation. 2D-only model has stronger semantic capability, but is less stable in complex scenarios such as those involving small objects. When the two are combined through full 2D–3D fusion, Single-Trial SR improves to 61.1%. These results support the interpretation of our paper: 3D primarily contributes fine-grained geometry and action-space constraints, while 2D mainly provides semantics and basic geometric cues. Robust manipulation requires their complementary fusion, rather than relying on either one alone.
>
> Q1. The current paper does not include a full ratio-sensitivity sweep, so we do not claim that the present ratio is globally optimal. At this stage, a more conservative and empirically grounded conclusion is that continuously exposing the model to multiple point-cloud sources during training brings stable benefits. In Setting 2, different point-cloud sources are continuously sampled during training, allowing the 3D encoder and fusion layers to learn geometric patterns that are less sensitive to the depth source. Table 4 also shows that, across three evaluation categories, hybrid training consistently outperforms or at least matches single-source training. Therefore, we would currently attribute the benefit to improved robustness brought by source diversity, rather than to the accidental optimality of one specific ratio. In the revision, we will list ratio scheduling/weighting and a systematic investigation of the mixing ratio as future work.
>
> Q2. The most common semantic failure is grasping the wrong object. For example, the instruction is to grasp object A, but the model instead grasps object B. We believe this is related to the weaker semantic capability of Concerto compared with the combination of DINOv2-SigLIP.

---

> > ### Author Rebuttal · Reviewer_wj8W · 2026-04-01
> >
> > I appreciate the authors’ detailed rebuttal. My concerns regarding the baseline methods and backbone have been addressed. I will raise my score to Weak Accept.

---

> > > ### Author Response · Authors · 2026-04-01
> > >
> > > We appreciate your recognition of our work and the effort you put into the rebuttal!

---

### Official Review · Reviewer_LHde · 2026-03-10

**Soundness:** 3
**Presentation:** 3
**Significance:** 3
**Originality:** 3
**Overall Recommendation:** 4
**Confidence:** 4

**Summary:**

This work presents a VLA pipeline that incorporates 3D features (point cloud) into VLA perception. The core innovation lies in a "2D-3D-2D" Patch Fusion architecture that re-projects extracted point cloud features back onto 2D image patches, applies geometric constraints while preserving the strong semantic representations of pre-trained foundation models. Also, it proposes a hybrid training strategy that integrates diverse point cloud sources, including simulator-generated, sensor-captured, and model-estimated (monocular depth) data. Experimental results show that ANY3D-VLA achieves generalization against viewpoint changes, occlusions, and scale variations. It also improves task success rates in complex real-world environments.

**Compliance With Llm Reviewing Policy:**

Affirmed.

**Final Justification:**

I thank the authors for their thorough response, and I have no further questions. I would like to keep my original rating **Weak Accept**.

**Key Questions For Authors:**

1. If using MDE models in VLA deployment, the latency might be high, especially for large MDE models like UniDepth or DepthAnything. Would the latency and computational overhead be feasible in a real-world deployment?

2. As mentioned above, however, there might be errors coming from using MDE (Monocular Depth Estimation) models, as the authors also mentioned in the paper. I am not sure if a hybrid training strategy could mitigate such errors or how such errors could be further mitigated.

3. This paper uses MDE models to get depth prediction. Since language instruction is also available, as investigated by previous works [R1, R2, R3, R4, R5], it might help MDE models to get better depth estimation. I think it’s a potential direction worth exploring or discussing.

[R1] Zeng, Z., Wang, D., Yang, F., Park, H., Soatto, S., Lao, D., & Wong, A. (2024). Wordepth: Variational language prior for monocular depth estimation. In Proceedings of the IEEE/CVF conference on computer vision and pattern recognition (pp. 9708-9719).

[R2] Zeng, Z., Wu, Y., Park, H., Wang, D., Yang, F., Soatto, S., ... & Wong, A. (2024). Rsa: Resolving scale ambiguities in monocular depth estimators through language descriptions. Advances in neural information processing systems, 37, 112684-112705.

[R3] Zhang J, Lu G. Vision-language embodiment for monocular depth estimation[C]//Proceedings of the Computer Vision and Pattern Recognition Conference. 2025: 29479-29489.

[R4] Zeng, Z., Ni, J., Wang, D., Rim, P., Chung, Y., Yang, F., ... & Wong, A. (2024). Iris: Integrating Language into Diffusion-based Monocular Depth Estimation. arXiv preprint arXiv:2411.16750.

[R5] Cui B, Huang Y, Bai L, et al. TR2M: Transferring Monocular Relative Depth to Metric Depth with Language Descriptions and Scale-Oriented Contrast[J]. arXiv preprint arXiv:2506.13387, 2025.

**Limitations:**

yes

**Strengths And Weaknesses:**

1. The proposed “2D-3D-2D” Patch Fusion seems novel. This architecture applies spatial geometric constraints to the VLA model by re-projecting native 3D point cloud features back onto 2D image patches while keeping semantic representations. However, combining 3D features (depth, point cloud, etc.) into VLA has already been explored by many other works, as also mentioned by the authors in the related works. I feel the novelty of this “patch fusion” is limited.

2. By training on a diverse mixture of simulator-generated, sensor-captured, and model-estimated data, the proposed Hybrid Point Cloud Training Strategy mitigates 3D data scarcity and gains the flexibility to operate in pure RGB environments using monocular depth estimation models. However, there might be errors coming from using MDE (Monocular Depth Estimation) models, as the authors also mentioned in the paper. I am not sure if a hybrid training strategy could mitigate such errors.

3. Experimental results show that ANY3D-VLA achieves generalization against viewpoint changes, occlusions, and scale variations. It also improves task success rates in complex real-world environments.

---

> ### Author Rebuttal · Authors · 2026-03-27
>
> Thank you for your positive review and recognition of our paper. We appreciate your detailed feedback and will address your concerns as follows.
>
> W1. Indeed, some prior works have already explored introducing depth or point clouds into VLAs. Our contribution is not merely to “use 3D features,” but also to reinforce the value of fusing native 3D representations with the corresponding 2D representations for VLA manipulation. First, under a strictly controlled setting, we conducted a pilot study comparing five observation/representation designs, and the results show that point cloud–2D patch fusion performs the best. Second, our method is not simple feature concatenation: we use a pretrained point encoder to model the native sparse 3D structure, then project the 3D features back to the corresponding 2D patches, and apply gated residual fusion so that 3D serves as a geometric correction to the strong pretrained 2D semantic representation, rather than directly replacing it.
>
> W2 & Q2. Regarding whether hybrid training can truly mitigate MDE errors, a more accurate way to state our claim is that it improves robustness to noise, scale mismatch, and geometric distortions across different depth sources. This is precisely the motivation behind Setting 2: by exposing the model during training to simulator-generated, sensor-captured, and model-estimated point clouds simultaneously, we encourage the 3D encoder and fusion layers to learn geometric patterns that are less sensitive to the depth source. The experimental results are consistent with this claim: in zero-shot real-world evaluation, Setting 2 outperforms Setting 1 under both RealSense and DA3 inference (57.5% vs. 55.0%, and 62.5% vs. 60.0%, respectively). In post-training, hybrid training also outperforms sensor-only training, reaching Task 1: 93.3% / Task 2: 86.7% under DA3 inference, compared with 80.0% / 60.0% for sensor-only training.
>
> Q1. Regarding latency and the feasibility of real-world deployment, we agree that this is an important issue. In our real-world experiments, we specifically compared different point-cloud estimators and ultimately selected Depth Anything 3 by jointly considering prediction quality and inference latency. We also use action chunking with a chunk size of 4, which helps amortize the perception overhead to some extent. In addition, compared with the higher-frequency ($\pi_{0.5}$), our model executes a larger motion per step, roughly 2–3× longer, so this operating frequency remains acceptable for real-robot deployment; representative videos are provided in the supplementary material. Our current target scenario is tabletop manipulation rather than very high-frequency closed-loop control, and we will make this latency–accuracy trade-off more explicit in the revised manuscript.
>
> Q3. We sincerely thank the reviewer for the suggestion regarding language-assisted monocular depth estimation. We agree that this is a very promising direction. In the current design, we intentionally keep the depth module task-agnostic, so that the VLA backbone can flexibly interface with point clouds from different sources without modifying the backbone itself. Accordingly, this work places greater emphasis on a modular interface: any stronger future depth/3D estimator can be directly plugged in as a replacement. At the same time, incorporating language priors into depth estimation indeed has the potential to further improve task-relevant scale disambiguation and target-centered depth quality. We will explicitly include this point in the future work discussion.

---

> > ### Author Rebuttal · Reviewer_LHde · 2026-03-31
> >
> > I thank the authors for their thorough response, and I have no further questions.

---

> > > ### Author Response · Authors · 2026-04-01
> > >
> > > We appreciate your recognition of our work and the effort you put into the rebuttal!

---

### Official Review · Reviewer_oNNX · 2026-03-12

**Soundness:** 3
**Presentation:** 3
**Significance:** 3
**Originality:** 2
**Overall Recommendation:** 4
**Confidence:** 3

**Summary:**

This paper explores the problem of incorporating 3D data in VLA design.  Despite the fact that robots are often equipped with 3D sensors (e.g. IR depth, LiDAR), the most successful VLAs are still almost entirely purely image-based. There are myriad explanations for this, but one important reason is that it is still an open question how best to represent the 3D information.  This is the investigation of this paper, and to address this Any3D-VLA is proposed, where the key contribution is the fusion of 3D features (encoding point clouds obtained by projecting depth maps) with 2D patch features (reprojecting the 3D encodings to the image plane).  This provides a simple way of integrating RGB-D inputs to RGB-based VLA training frameworks.

To deal with limited 3D data and generalization challenges, a hybrid training is proposed, where simulator or monocular depth prediction point clouds are interleaved.  Any3D-VLA outperforms baseline models on simulated and real-world benchmarks.

**Compliance With Llm Reviewing Policy:**

Affirmed.

**Final Justification:**

I believe there are fundamental limitations on the efficiency of the proposed approach, which is a critical point for the application of VLAs. Furthermore, this issue is closely connected to the heuristic nature of the method (voxelization/pt.cloud reduction method which is already extreme), and utilizing monocular depth models which introduces an unavoidable overhead.  My opinion on both these points was confirmed through the rebuttal.  Thus, I will maintain my original rating of Weak Accept.

**Key Questions For Authors:**

Please see my questions above in the Strengths and Weaknesses section.  Additional questions are below:

- In Table 3 why are \pi_0.5, GraspVLA and SpatialVLA performing so poorly relative to Any3D, despite undergoing the same post-training regimen?  The gap in performance is much bigger than for other benchmarks. Is there an explanation of why these challenges in particular would be so difficult for the baselines?

- There are many simulation datasets in robotics environments that could be used for training VLAs, and many might have simulated depth as well.  What is the motivation for creating a new dataset?  Also, will the dataset be released and free to use for research purposes?

**Limitations:**

yes

**Strengths And Weaknesses:**

**Strengths**
- This paper investigates an important problem, as integrating 3D information to make VLAs “3D-aware” is an open challenge.
- The proposed techniques are intuitive and well motivated, and extensively evaluated through experiments and ablations.
- Hybrid point cloud training is a simple and effective solution to deal with limited 3D training data and improve generalization.

**Weaknesses**
- More discussion is needed surrounding the performance vs speed/latency tradeoffs (in robotics settings this is unavoidable).  The latency discussion feels hidden in the appendix, rather than treated as a first-class metric.  Can a fair comparison be provided with the baseline models.  I imagine running a monocular depth and 3D encoders (concerto) is much more compute-heavy than the baselines.  Given that there is already heavy 3D compression (voxelization, selecting 1 pt per cell), there might not be a simple way to achieve further efficiency gains, so any latency issues may be fundamental limitations of the idea.
- On some datasets the performance is *significantly* worse when compared to some prior works (Appendix J, Libero/Calvin). This is attributed to SpatialVLA  and \pi_{0.5} being trained on richer and more diverse datasets.  When making a fundamental representation change, I think it is important to know if there are regressions along dimensions where 3D is potentially not helpful.  Is there a way to design a fair(er) experiment on these datasets?  Perhaps a baseline VLA (e.g. one of the models from Table 1 that doesn’t use depth maps) can be trained on the proposed dataset, and its generalization performance to Libero/Calvin would provide a good reference point for the full Any3D model.
- The depth data (pre)processing is heuristic-driven.  It seems like some of these might hinder generalization. For example cropping the background might not even make sense for different robot scenarios.  Also, one wonders how sensitive the performance is to these processes (e.g. performance vs voxel size).  For the RGB-only pipelines the image encoders are established and well-studied so there is less uncertainty there.
- The idea of fusing 2D and 3D modalities for training the encoder is not entirely novel.  In fact the Concerto model used for 3D encoder here is itself pretrained by fusing 2D patch features and point cloud. The proposed approach (concatenating 3D+2D tokens + MLP gating) is a minor variation to Concerto.  There are also other related fusion ideas in the literature:
    - ODIN: A Single Model for 2D and 3D Segmentation (https://arxiv.org/pdf/2401.02416)
    - Bidirectional Projection Network for Cross Dimension Scene Understanding (https://arxiv.org/pdf/2103.14326)
    - Rgb-D Fusion For Point-Cloud-Based 3d Human Pose Estimation (ICIP 2021)

At a high level, the implementation details of the proposed approach is a specific instantiation of the same conceptual fusion idea.


I look forward to reading the author response and discussing with the other reviewers.

---

> ### Author Rebuttal · Authors · 2026-03-28
>
> Thank you for your positive review and recognition of our paper. We appreciate your detailed feedback and will address your concerns as follows.
>
> W1. We agree that latency should be treated as a first-class metric in the main paper rather than being relegated to the appendix. In the revised manuscript, we will add the following efficiency–performance trade-off:
>
> || Depth source | Inference speed | Point cloud size (after preprocessing) ||
> | ---------------------- | ---------------- | --------------: | -------------------------------------: | --------------------------------- |
> | 2D baseline (GraspVLA) | – | 3.0 FPS | – | Fastest |
> | Real-world | RealSense | 2.0 FPS | ~3k–8k points | Faster |
> | Real-world | Depth Anything 3 | 1.7 FPS | ~3k–8k points | Better accuracy–latency trade-off |
>
> The raw point cloud is first processed through cropping and 3D compression, reducing it from approximately 30k–60k points to about 3k–8k points. In actual deployment, we ultimately chose DA3 because it provides a better balance between accuracy and latency.
>
> W2. Structurally, GraspVLA is the version of Any3D-VLA without using depth information. In our experimental comparison, the two models are trained under exactly the same pipeline and data setting (pretraining + LIBERO/CALVIN), making GraspVLA a fair 2D baseline. Any3D-VLA improves over GraspVLA by 13.9% on LIBERO and by 0.71 in average length on CALVIN. This indicates that the performance gain comes from the 3D representation itself. We do not observe systematic degradation relative to the corresponding 2D baseline.
>
> W3. The preprocessing of images and point clouds can be flexibly adapted to the specific robot scenario, including whether background cropping is appropriate. The voxel size of 1cm was also chosen as a trade-off between manipulation precision and efficiency. We will add sensitivity to preprocessing choices as a direction for future work in the revised manuscript. Moreover, cropping and voxelization are engineering design choices rather than the core conceptual claim of the paper. Cropping is mainly used to reduce background interference while preserving RGB–geometry alignment, whereas voxelization is used to improve computational efficiency.
>
> W4.
> We agree that the general idea of fusing 2D and 3D information is not new, and we thank the reviewer for pointing out relevant works such as ODIN. We will further strengthen the related work discussion in the revision. Our contribution is not to claim the first 2D–3D fusion approach, but rather to introduce a 2D–3D fusion interface tailored for pretrained VLAs, enabling native point-cloud representations to be effectively injected into an existing 2D VLA. Specifically, we first encode the native point cloud in the camera coordinate frame, then align 3D features to ViT patches through patch-wise reprojection, and finally inject them into 2D tokens using residual correction, together with an empty-token mechanism to handle sparse patches. The goal of this design is to introduce 3D geometric corrections while preserving as much of the pretrained 2D semantic representation as possible. This differs from approaches that treat 3D as merely a parallel branch, directly fuse features after simple concatenation, or replace 2D tokens altogether. Our method is not a simple variant of Concerto. Concerto serves here as a pretrained 3D encoder, whereas the focus of this paper is how to connect native 3D features to a pretrained VLA in a patch-aligned and plug-and-play manner, and to validate its effectiveness in manipulation tasks.
>
> Q1. These two post-training tasks both rely on precise 3D alignment, and each task contains only 100 demonstrations. RGB-only models ($\pi_{0.5}$, GraspVLA) are more fragile when manipulating transparent and reflective objects (the condiment cup). In addition, these tasks do not provide bounding box annotations, making precise localization more challenging for GraspVLA. SpatialVLA takes as input relatively noisy RealSense depth, while its pretraining is conducted on near-perfect depth, leading to performance drop. Overall, these baseline models are prone to lateral grasp-position drift, localization shifts under occlusion, and failure in target localization under appearance-deprived conditions. Such issues are further amplified in new tasks that require fine-grained geometric alignment.
>
> Q2.
> There do exist some simulation datasets that provide depth; however, their scale is significantly smaller than ours, and they lack the diversity in object categories, spatial configurations, and backgrounds present in our dataset. A large-scale RGBD simulation dataset is itself a meaningful contribution, as it can enhance the generalization capability of VLAs and help reduce the sim-to-real gap. The current submission already states that the code will be released anonymously; upon acceptance, we will also open-source a portion of the dataset for free use by the research community.

---

> > ### Author Rebuttal · Reviewer_oNNX · 2026-04-03
> >
> > Thanks for responding to each of my listed weaknesses and questions in detail.  I now have sufficient information to finalize my review.

---

> > > ### Author Response · Authors · 2026-04-04
> > >
> > > We appreciate your recognition of our work and the effort you put into the rebuttal!

---

### Decision · Program_Chairs · 2026-04-30

**Decision:**

Accept (regular)

**Comment:**

The paper considers the important issue of bringing 3D information into vision-language-action models. The proposed method, Any3D-VLA, is a modular framework. It adds point cloud features into pretrained VLA pipelines, via a 2D–3D–2D patch fusion mechanism and a hybrid training strategy. The latter makes the joint use of simulator, sensor, and monocular-depth point clouds. The reviewers agree that the problem is relevant, the approach is technically sound, and the experiments are convincing in that scenarios that can be expected to benefit from 3D geometry do indeed so. Special appreciation was expressed for the real-world experiments. Also some concerns were raised. Examples are limited novelty when compared to earlier 2D–3D fusion work, considerations related to efficiency, and baseline comparisons. The rebuttal positioned Any3D-VLA as a plug-in option for pretrained VLAs,. It also provided further runtime info and defended the experiment design. Even if the novelty is moderate and efficiency remains a consideration, the paper proposes a useful step towards 3D-aware VLAs. Therefore, and of course based on the reviewers' judgement, I support acceptance.